# Dehydration alters behavioral thermoregulation and the geography of climatic vulnerability in two Amazonian lizards

**Agustín Camacho** [1,2]*, **Tuliana O. Brunes**[3], **Miguel Trefaut Rodrigues**[3]

**1** Departamento de Fisiologia, Instituto de Biociências, Universidade de São Paulo, São Paulo, Brazil,
**2** Departmento de Ecología Evolutiva, Estación Biológica de Doñana, Sevilla, España, **3** Departmento de Zoologia, Instituto de Biociências, Universidade de São Paulo, São Paulo, Brazil

\* agus.camacho@gmail.com

**Data Availability Statement:** All relevant data are within the paper and its Supporting Information files.

## Abstract

High temperatures and low water availability often strike organisms concomitantly. Observing how organisms behaviorally thermohydroregulate may help us to better understand their climatic vulnerability. This is especially important for tropical forest lizards, species that are purportedly under greater climatic risk. Here, we observed the influence of hydration level on the Voluntary Thermal Maximum (VTmax) in two small Amazonian lizard species: *Loxopholis ferreirai* (semiaquatic and scansorial) and *Loxopholis percarinatum* (leaf litter parthenogenetic dweller), accounting for several potential confounding factors (handling, body mass, starting temperature and heating rate). Next, we used two modeling approaches (simple mapping of thermal margins and NicheMapR) to compare the effects of dehydration, decrease in precipitation, ability to burrow, and tree cover availability, on geographic models of climatic vulnerability. We found that VTmax decreased with dehydration, starting temperature, and heating rates in both species. The two modeling approaches showed that dehydration may alter the expected intensity, extent, and duration of perceived thermal risk across the Amazon basin for these forest lizards. Based on our results and previous studies, we identify new evidence needed to better understand thermohydroregulation and to model the geography of climatic risk using the VTmax.

## Introduction

State-of-the-art protocols to map species' climatic vulnerability rely on estimations of thermal tolerance, thermoregulatory behavior, and water loss [1–3]. Due to data availability, these traits have been mostly considered as fixed for species [1–3] despite it being well-known that the physiological performance of ectothermic animals is sensitive to temperature and hydration level. For example, dehydrated reptiles and amphibians often exhibit lower critical thermal maxima (*i.e.*, a high body temperature which, if surpassed, would impede the individual's locomotion [4–6]) and lower optimal temperatures for sprint speed [7]. As body temperature rises, evaporative body water losses can increase exponentially [8], like metabolic costs [9]. Also,

**Funding:** Agustín Camacho:CAPES/PNPD 001. MSCA:897901 Tuliana O. Brunes: FAPESP 2016/03146-4 Miguel Trefaut Rodrigues: FAPESP: 2011/50146-6. CNPQ: 301778/2015-9. The funders had no role in study design, data collection and analysis, decision to publish, or preparation of the manuscript.

**Competing interests:** The authors have declared that no competing interests exist.

while dehydration may alter ectotherms' growth rates [10], thermal and hydric stress combined may decrease reproduction and survival [11].

When ectothermic animals, such as lizards, exhibit thermoregulatory behaviors, they likely integrate information on their hydration level (i.e. they thermohydroregulate) [12]. For instance, dehydrated lizards required exposure to higher temperatures to induce the onset of water-demanding thermoregulatory behaviors, such as panting or urinating [13–18]. Notwithstanding, these two behaviors may represent emergency responses (reviewed in Tattersall et al., [19]). When dehydrated and maintained in laboratory thermal gradients, amphibians and reptiles may exhibit somewhat lower mean body temperatures [5, 11, 20–23]. This preference for lower temperatures is often associated with the use of sheltering microhabitats [10, 22], potentially making hydration levels a stronger determinant of habitat use and activity time than temperature [24, 25]. Likewise, studying species' responses to evade both thermal and water stress should help us understand climatic restrictions on their activity and distribution.

The voluntary thermal maximum (VTmax) is a thermoregulatory trait shown to integrate both thermal and water levels in frogs [21]. This trait is key for mechanistic models of ectotherms' daily activity and climatic vulnerability [26]. When measured in the field, this is a temperature that forces animals to seek retreat [4], when obtained experimentally, it can be considered the maximum body temperature an animal accepts before exhibiting a clear evasion behavior from a heat source [27]. Both intra and interspecific variations in the VTmax are scarcely explored. So far, studies on active thermoregulator species (*e.g.*, Jarrovi's lizard) have shown the VTmax can be geographically invariant across populations separated by millions of years [28] and unresponsive to changes in oxygen concentration [29], or to typical experimental conditions of thermal tolerance assays, like heating rates and starting temperatures (*i.e.*, the temperature at which the heating starts [27]). Notwithstanding, typical thermoconformers (*e.g.*, the lizard *Hemiergis peronii*) do respond to them, making them thus important factors to account for, as we did. Finally, since individuals may warm up in a range of conditions, it becomes important to document how they would avoid stressful temperatures in different situations. Previous studies in desert lizards have found no effects of dehydration and starvation on the VTmax [14], although this parameter may vary after the injection of saline solutions in lizards, a different way to alter plasma osmolality levels.

By combining thermal tolerance traits, like the VTmax, with environmental temperatures, researchers can estimate the exposure of a species to thermal risk [30]. Exposure to the VTmax can induce function loss and kill within four hours. Yet, two desert species have been observed to survive to exposures beyond 24 h (*e.g.*, *Chalcides ocellatus* and *Tarentola annularis*) [27, 31]. This means that whenever the minimum temperature available for a population reaches its VTmax, many individuals would experience stressful temperatures at their refugia. If after reaching their VTmax, the individuals leave their refugium, they could be exposed to much greater and likely lethal temperatures under the sun [32]. Likewise, individuals sheltered in hotter refugia might be overwhelmed by temperatures that exceed thermal limits.

The magnitude of thermal risk, often called thermal safety margins, represents how much a species' upper thermal tolerance exceeds environmental temperatures [33]. Although upper thermal tolerance has often been estimated as the critical thermal maximum [32], the VTmax is becoming increasingly used to identify climatic restrictions on populations [34]. Following, the exposure to thermal risk can be represented by how much of a species' geographic range is affected by negative margins, or for how long stressful temperatures are present at each site of that range. The latter is particularly relevant when using a trait that does not kill immediately, like the VTmax.

In thermal risk analyses, air temperature measured in the shade has been used to estimate the minimum temperature available for dry-skinned ectotherms (Ex. squamates, terrestrial

arthropods) [1, 33, 35]. Yet, some small ectotherms might still bury themselves in moist micro-habitats and experience lower maximum temperatures [36], making it important to evaluate how changes in thermal tolerance (*e.g.*, in the VTmax) and in the availability of such microhabitats might alter estimates of thermal vulnerability. Using mechanistic models of heat and water exchange between individuals and their environment (*e.g.*, the Niche Mapper [37]), researchers can simulate the effects of changes in the availability of sheltered microhabitats (*e.g.*, underground or under tree cover) and other conditions (environmental water, precipitation). This allows for comparing the relative importance of changes in expected vulnerability due to changes in the VTmax due to dehydration with changes in other factors, such as the availability of water itself. In this context, the amount of annual precipitation arises as a climatic condition capable of affecting the interaction between voluntary thermal tolerance, dehydration, and thermal safety. Rain provides water for animals and also dampens daily thermal oscillations [36] through cloud cover and later evaporation of soil water. Therefore, changes in the geographic distribution of rain should be capable of changing the geography of predicted thermal vulnerability. Still, there is a lack of studies that compare the effects of changes in thermohydroregulation and rain on the geographic distribution of climatic vulnerability, especially in the tropics, where many species are considered thermoconformers and potentially subject to higher thermal risk [33, 38–40].

Herein, during a faunal inventory in the Amazon basin, we tested whether dehydration lowers the VTmax in two small thermoconformer lizard species from central Amazonia (*Loxopholis ferreirai* and *Loxopholis percarinatum*). Later, we illustrated how changes in the VTmax, the available soil depth for burying, tree cover, and precipitation might alter the geography of perceived thermal risk for the more widely distributed *L. percarinatum*, across the Amazonian Basin.

## Results

### Effects of treatments on body mass

Hydration/dehydration treatments affected lizards mass as expected, leading to a statistically significant decrease in body mass in the second (dehydrated) trial, compared to the fresh trial (difference with fresh mass: -0.075 g, SD: 0.006, DF: 54, t:-12.244, p-value: 2e-16) and a recovery in the third (re-hydrated) trial (difference with fresh mass: -0.003 g, SD: 0.006, DF: 54, t = -0.553,p-value: 0.582). Fig 1 shows the variation in mass for both species across treatments, separately. Statistical power was very high for this test (90% CI: 96.38–100%).

### Effects of hydration level and experimental factors on the VTmax

During our model selection process, all models had acceptable statistical power (171–98%, mostly over 80%) (S1A Table in S1 File). Among the fitted models, only starting temperature (range: 22.5~29.8˚C) and heating rate (range: 0.12–3.7˚C/min) improved the model's fit after including hydration level (S1A Table in S1 File). Thus, neither trial nor body mass was included in the final model for any of the two species (Fig 2).

For *Loxopholis ferreirai*, the VTmax ranged from 27.02 to 36.6˚C (N = 9) and was significantly affected by hydration level, heating rate, and starting temperatures (all p<0.05). VTmax increased at an average of 0.12˚C per percentual unit in hydration level (95%CI: 0.01–0.22). For hydrated lizards (>95% hydration level), the median VTmax was 32.8˚C and the median body mass was 0.694. For dehydrated lizards (<95%hydration level) the median VTmax was 29.2˚C and the median body mass was 0. 671. The VTmax increased an average of 0.77˚C per ˚C increase in the starting temperature (95%CI: 0.40–0.77), and also increased 2.3˚C per

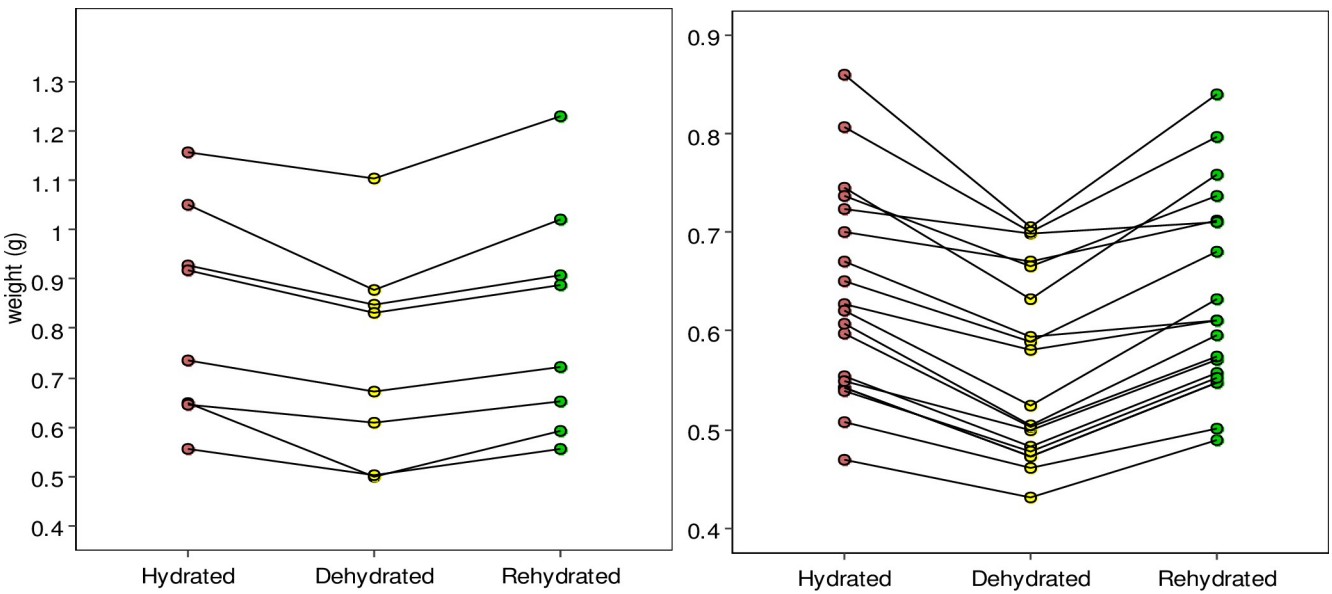

**Fig 1. Changes in body weight as a result of overnight dehydration and rehydration treatments in *Loxopholis ferreirai* (left) and *L. percarinatum* (right) lizards.**

degree/minute increase in heating rate (95%CI: 0.50–4.2), respectively (Fig 2 and S1 Table in S1 File). The confidence interval of statistical power for that model was (83.60–95.80%).

Among *Loxopholis percarinatum*, the VTmax ranged from 24.1 to 33.3˚C (N = 19) and was again significantly affected by hydration level, heating rate, and starting temperatures. The VTmax increased an average of 0.12˚C per percentual unit in hydration level (95%CI: 0.05–0.18). For hydrated lizards (>95% hydration level), the median VTmax was 31.2˚C and the median body mass was 0.628 g. For dehydrated lizards (<95% hydration level), the median

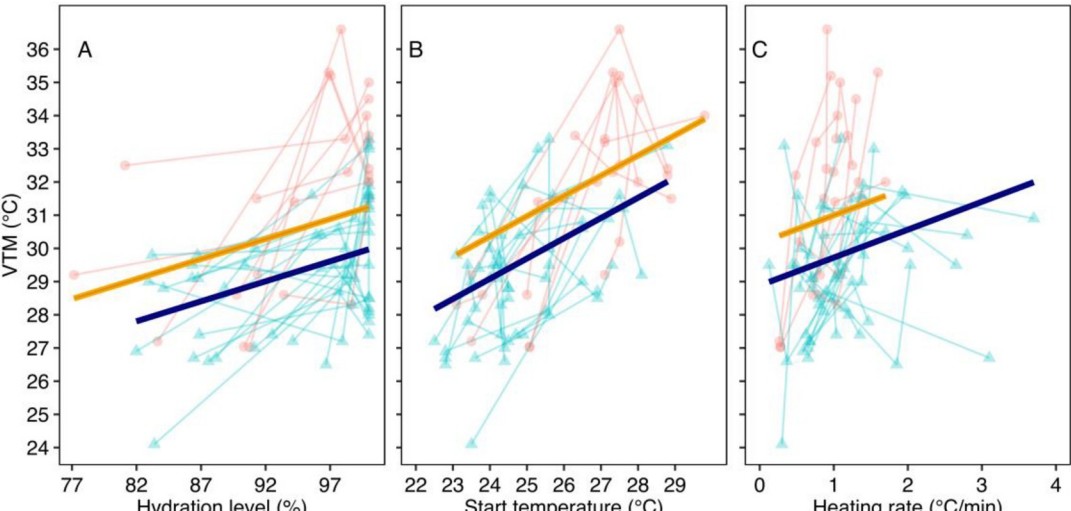

**Fig 2.** Repeated observations of the VTmax in relation to individuals' hydration level (A) start temperature (B) and heating rate (C), measured in *Loxopholis ferreirai* (Orange circles) and *L. percarinatum* (blue triangles) lizards. Points indicate observations, and bars represent predicted values by the best-supported model, which includes the three factors (see methods). Graphs plotting residual values can be viewed in S1 Fig in S1 File.

VTmax was 28.8˚C and the median body mass was 0. 525 g. Increasing starting temperature and heating rate again significantly increased the lizards' VTmax. The VTmax increased an average of 0.41˚C per ˚C increase in the starting temperature (95%CI: 0.12–0.70), and also increased 0.58˚C per degree/minute increased in heating rates (95%CI: 0.08–1.09), respectively (Fig 2). Results after applying the *rankit* function for the normalization of residuals led to similar conclusions for this species (S2 Table in S1 File). The confidence interval of statistical power for that model was (96.38, 100.0%).

## Maps of thermal risk across the Amazon basin

Our simple mapping approach estimated that, for hydrated *L. percarinatum*, only 6% of known populations would be currently under negative thermal margins (Fig 3A). However, thermal margins became more negative for dehydrated lizards, leading to 87% of known populations reaching negative thermal margins if they were dehydrated below 95% of their fully hydrated mass (Fig 3B).

Concerning the 64 NicheMapR models generated, panels C-F in Fig 3 illustrate a sample of the most important effects. The extent, the total duration of thermally stressful events, and the percentage of known lizard populations affected by them, were several orders of magnitude more increased by decreases in the VTmax, associated with dehydration, than by any other parameter. The amount of rain and the percentage of tree cover follow, in mixed order of importance depending on the metric used. While changing the amount of rain was more important for determining the number of known populations affected by negative margins, the amount of tree cover had a larger effect on the extent and duration of thermal stress across the Amazon basin (Table 1, S4 and S5 Tables in S1 File provide full summaries). Neither the ability to bury down to 10 cm nor changes in mass associated with dehydration did not observably affect our thermal risk metrics.

## Discussion

Although they are often considered thermoconformers [14, 38, 40], we showed that rainforest leaf litter lizards can adjust their VTmax with respect to hydration level, and also integrate heating rates and starting temperatures. Different from typical laboratory conditions, individuals in natural conditions may heat up in a wide range of circumstances. Because of this, even if our experimental set up is not identical to a natural warming process, our results suggest that these lizards should respond to these factors in the wild. Although the sampling sizes could be considered somewhat low, given the homogeneous responses observed independently in almost every treated individual in two different species, we are confident that the observed effects reflect a biological mechanism. Additionally, VTmax has responded similarly to starting temperatures and heating rates in other thermoconformer lizards [27], in the bullfrog [21], and in leaf-cutting ants [41], but not in active-thermoregulators (*e.g.*, phrynosomatid lizards [27] herein). Thus, the VTmax may be regulated by these factors with taxon-specific importance among ectotherms. Considering that the *Loxopholis* genus has inhabited Amazonian forests for at least 10 million years [42], their capacity for behavioral thermohydroregulation does not seem residual from a recent past in open habitats.

Future analyses of factors affecting the VTmax of ectotherms should help us uncover intra and interspecific patterns in thermohydroregulation. Intraspecifically, individuals should exhibit behavioral responses to maintain optimal thermal and hydration levels, except in the rare cases when individuals allow their inner environment to fluctuate largely [43]. These should include behavioral responses, such as changes in posture, microhabitat use, and/or activity time [12]. The existing evidence suggests that the VTmax response can integrate both

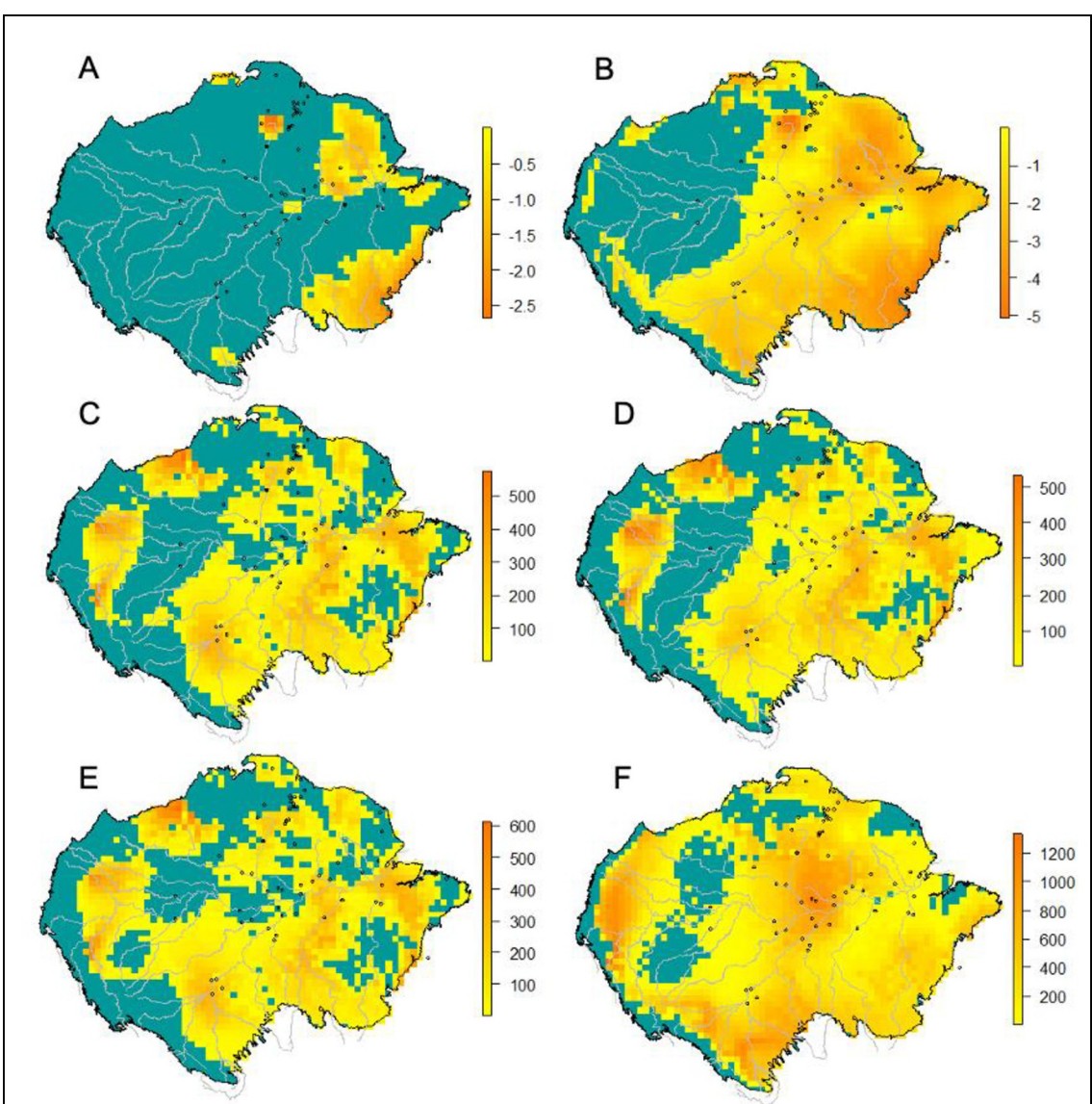

**Fig 3. Map of thermal risk variation, based on the response of lizards to dehydration for *Loxopholis percarinatum* across the amazon basin.** Points represent known populations of these lizards. Pixels in yellow to orange represent areas reaching negative thermal margins, while green represents areas where thermal margins remained positive. A and B represent the difference between the median VTmax of hydrated (A) or dehydrated (B) lizards and the mean of the yearly maximum air temperatures at any pixel. In C to F, the same colors show the number of hours that body temperatures exceeded the VTmax for hydrated and dehydrated lizards when allowed to seek refuge down to 10 cm under wet leaf litter in the shade. C represents a "best" situation, with hydrated lizards (VTmax = 31.2˚C), 100% tree cover, full rain, and ability to burrow down to 10 cm. Instead, D to F show the expected effects of altering some of these parameters. D represents thermal risk after a 50% decrease in rainfall, E shows risk after a decrease of 20% in tree cover, and F shows the risk for lizards with VTmax = 28.8˚C, The median VTmax for *L. percarinatum* under 95% of their maximum hydration level.

temperature and hydration level, and induces at least two of these response types [26, 44, 45]. Therefore, the lower VTmax in dehydrated individuals might persuade them to progressively move to cooler microhabitats, as daily thermal fluctuations, often more intense in drier environments, become more likely to overcome their lowered VTmaxs. Such a process may explain several experimental and field observations of microhabitat use during natural droughts of temperate and tropical lizards [10, 24, 45] (herein). If air temperature in the shade reaches the

**Table 1. Effects of changing different parameters on the extent (number of sites across the Amazon basin), duration (number of yearly hours over the VTmax), and number of known populations affected by thermally stressful events for *L. percarinatum* across the Amazon's basin.** Parameters: Mass = body mass, VTmax = voluntary thermal maximum, Moist_surface = percentage of surrounding surface that can evaporate water, Maxdepth = maximum depth to which a lizard can dig, Rain = amount of rain falling, Maxshade = Maximum percentage of tree cover that the lizard has access to. See methods to consult the levels compared among factors. See S3-S5 Tables in S1 File for full results.

| | Extent | | Duration | | Populations | |
|---|---|---|---|---|---|---|
| | t-value | p-value | t-value | p-value | t-value | p-value |
| (Intercept) | 39.16 | 0.00 | 22.06 | 0.00 | -44.14 | 0.00 |
| Mass_0.628g | -0.13 | 0.90 | -0.02 | 0.98 | -0.09 | 0.93 |
| VTmax_31.2C | -32.45 | 0.00 | -21.56 | 0.00 | -52.27 | 0.00 |
| Moist_surface_100% | -0.09 | 0.93 | 0.04 | 0.97 | 0.28 | 0.78 |
| Maxdepth_10cm | -0.03 | 0.98 | -0.03 | 0.98 | -0.09 | 0.93 |
| Rain_full | -5.34 | 0.00 | 1.44 | 0.16 | -9.68 | 0.00 |
| Maxshade_100% | -11.42 | 0.00 | -7.54 | 0.00 | -1.93 | 0.06 |

decreased VTmax more often, sheltering, and potentially stopping activity, might be expected. By achieving lower body temperature, ectothermic animals decrease their rates of body water loss and energy consumption [46, 47, respectively], although lizards may decrease body water loss in other ways, too (reviewed in Pirtle et al., [48]). However, the absolute performance of essential traits, like locomotion, hearing, digestive and immune efficiency, and the rates of metabolic water production (ex. in desert-adapted reptiles), can decline as well (reviewed in [9, 12, 49]). These multiple effects justify why many ectotherms resort to sheltered inactivity (*i.e.*, aestivation) under extreme heat and drought. Yet, this response may strongly decrease reproductive output during particularly hot and dry years [11]. However, as stated before [12], identifying interspecific patterns in all these steps requires further research on the relationships between behavioral thermohydroregulation and population growth.

Across species, stronger effects of hydration on the VTmax could be initially expected for species from hotter and drier habitats. Coincidently, Heatwole and Firth [50] already commented on *Amphibolorus* lizards exhibiting lower VTmax after summer and that these changes were lower for species from more humid environments. They attributed these changes to internal rhythms or rhythms of the external environment. However, experimental dehydration did not lower the VTmax of the desert iguana [14], nor the maximum of body temperatures of temperate Lacertidae species studied in thermal gradients [20]. Still, desert iguanas did alter their VTmax in response to injections of concentrated saline solutions, leading to osmotic imbalance. Yet, desert iguanas have especially efficient ways to keep the ionic balance under heavy loses of body water, like salt glands, renal secretions, and water stored in their thick tails [14, 51, 52]. Moreover, these lizards often resort to panting when thermally stressed, which aids in cooling down the head and keeping the body warm [53, 54] without necessarily retreating from the heat source. This behavior is common among lizards from deserts and temperate regions, and it has been shown that panting temperatures may increase when the hydration level drops [13, 55]. Maintaining a high VTmax at the cost of dehydration might help lizards (and other organisms) behave more flexibly. This could allow them to maintain activity, with the resultant water ingestion, if available [56]. We hypothesize that such a strategy might be especially effective for organisms with special water reserves, such as the bladder (ex. the Gila monster [18]), and/or for species with low specific water loss rates (*e.g.*, large desert lizards or tortoises [43, 47, 57]). In turn, for smaller species, the challenges of keeping at relatively high temperatures and hold their hydration level should be especially important [58]. For example, the lizards studied herein (range: 0.21 to 1.23 g). Notwithstanding, the commonly shorter life span of smaller reptiles [59] might impose selective pressures on keeping activity levels high.

Thus, while more comparative studies are needed before reaching to patterns in thermohydroregulation strategies, the VTmax appears as a sensitive and easy parameter to explore such patterns.

The few ectothermic animals where thermohydroregulation has been studied show different behavioral strategies. While tropical lizards (herein), amphibians (frogs and salamanders, [21]), and even ants [60] lowered their VTmax when dehydrated, preliminary observations of desert lizards and temperate lizard species indicate that they maintained their maximal body temperatures either after experimental saline injections, or during dehydrating processes [15, 16, 23, 43, 47, 57]. Still, the mean preferred temperature of studied temperate species decreased. The data at hand is still very limited but allows hypothesizing that dehydration might induce less important decreases in the VTmax in species for whom keeping a high body temperature is necessary (*e.g.*, due to the unavailability of refuges or short reproductive seasons).

In contrast, for ectothermic animals from tropical rainforest, and particularly for parthenogenetic lizards, like *L. percarinatum*, which reproduce almost year-round [61], avoiding high temperatures by waiting for the rain in a relatively cool refuge should be less problematic. This strategy should help in avoiding the critical increases in thermal risk that may ensue decreases in rain frequency and more long-term reductions in shade, as suggested by our geographic models (Fig 3). Microhabitat use might also induce interspecific variation in the coupling of the VTmax and hydration level. For example, the greater reactivity of *L. ferreirai* might be expected for a species that can dehydrate and rehydrate faster. This agrees with the ecology of *L. ferreirai*, which climbs branches of flooded forests [62], where the environment is more desiccating than at the forest's leaf litter, but grants quick water access. Yet, although the rates of change in VTmax per unit in hydration level were 20% on average more intense for *L. ferreirai* compared to the *L. percarinatum* complex, this proportion represents a fraction of a degree in an absolute sense. In any case, two species comparisons do not allow to infer causality on the divergences found among them [63], nor is the purpose of this study. Finally, the fact of *L. percarinatum* be a complex, composed of 2n and 3n lineages with cryptic diversity, and a not yet described bisexual species [64], makes even harder to compare it with *L. ferreirai*. Future studies might detect more subtle differences in the VTmax, unobservable at present, associated with ploidy or parthenogenetic reproduction, as seen for other traits among parthenogenetic lineages of geckos [65]. Actually, in light of such genetic diversity within our study lizards (see methods), our results reaffirm the idea of a general effect of dehydration among these lineages, lowering the VTmax of individual lizards.

Based on the discussed lines of evidence, we propose that the VTmax of lizards, and possibly other ectothermic species, might be relatively more reactive to dehydration when the measured species: 1) presents less effective mechanisms for controlling plasma osmolality (e.g., salt glands), 2) lacks important water reservoirs in the body (*i.e.*, very thick tails, large bodies), 3) loses body water faster, 4) has easier access to cool and humid microhabitats, 5) inhabits places where growth rates do not need to be maximized (*i.e.*, thermally stable and humid), or 6) presents asexual reproduction. Testing these hypotheses in species of different habitats, will help add dehydration-induced variation in thermal tolerance to models of climatic risk.

Although our models should only be used for illustrative purposes, the large effects that VTmax changes had on the geography of thermal risk highlights the need of accounting for thermohydroregulation when modeling climatic vulnerability [25, 66]. The VTmax offers a higher sensitivity to detect thermal stress than the thermal limits: some desert lizards may show one to three degrees between the VTmax and the CTmax [27, 31], but this difference may reach as high as five degrees in *L. percarinatum* [67]. However, interspecific variation in the magnitude of this thermal fear zone (*i.e.*, between the VTmax and the CTmax), and the

lack of published data on it, might be seen as a problem in comparing thermal risk among species. While extremely useful for understanding thermal risk [68], measuring the time to heat shock across vertebrates raises important ethical and practical problems. The scarce published data on lizards [reviewed in Curry-Lindhal [27, 31], makes us tentatively suggest that whichever site at which the minimum available temperatures (*i.e.*, at species' shelters) overcome the VTmax for over three hours should induce enough stress to make a population vulnerable. Processes arising from failed thermoregulation at suboptimal refuges, and the concentration of individuals at thermal refuges, dehydration, and negative intra and interspecific interactions could likely harm the stressed population in such a situation. Our maps (Fig 3C–3F) show large regions across the Amazon basin, in which hundreds of hours overpass the VTmax of our study species.

The low congruence between the risk maps produced by our two approaches was expected. Apart from measuring relatively different variables (*e.g.*, intensity versus duration of exposure), the NicheMapR uses many more parameters than the simple mapping approach, and some of them impacted strongly the geography of thermal risk (*i.e.*, precipitation and tree cover), although less than changes in the VTmax. NicheMapR also allows accounting for the ability of lizards to bury, down to 10 cm in our case. This option, while able to provide large thermal protection for related lizards in dry tropical habitats [69], showed relatively low importance among the factors compared herein. Lower soil moisture has been found to either decrease [36] or increase [70] the intensity of underground thermal gradients. Such context-dependency, or perhaps a general excess of precipitation, might explain why the NichemappeR parameter "percentage of evaporating surface", did not have clear effects on the resultant thermal risk maps. In any case, further studies are needed to better understand thermoregulation options offered by burying behavior and soil moisture. We interpret the combined output of our models as clear warnings that water shortages across the amazon basin will dramatically increase thermal risk for dehydrated leaf litter lizards, but that the output may largely depend on local vegetational responses to changes in climate.

In conclusion, data on patterns of species' thermohydroregulation, aestivation, and field evaluations of presence in regions were models give opposing predictions, are badly needed before we can predict the impacts of changes in the hydrothermal environment (*e.g.*, through climate change, forest fires, or deforestation) on animal populations. Given that, researchers should not spare efforts to raise this information.

## Methods

### Species accounts and obtention of specimens

Our two studied species belong to the Gymnophthalmidae, a family older than 50 MY and mostly composed of leaf litter forest species [42]. *Loxopholis ferreirai*, is a semiaquatic and scansorial lizard species that lives over and within logs in flooded "Igapó" forests across the course of the Rio Negro and tributaries [62]. Our second species is *Loxopholis percarinatum* a parthenogenetic leaf litter species, typical of central Amazonian "Terra firme" forests [64]. As inhabitants of some of the most thermally stable and humid microhabitats on land (*i.e.*, Amazonian leaf litter and *Igapó*), these lizards have been regarded as thermoconformers, namely, behaviorally insensitive to changes in the thermal environment [2, 33, 38].

Our dataset consists of nine adult *L. ferreirai* from both sexes and 19 *L. percarinatum*, including only females (the latter is a parthenogenetic species' complex). Sizes for both species varied from almost newborn size to adult (see individuals' descriptions in the S1 File). All were collected during a faunal inventory across the Rio Negro, in April 2018 under the license number (SISBIO 30309–11). Thus, *L. ferreirai* come from two *Igapós* near to Boa Vista. *Igapós* are

forests typical from Amazonian rivers, which experience seasonal floods. In turn, individuals of *L. percarinatum* were collected at four *terra firme* forests, located in the margins of the Rio Negro, across 500 km from the Parque Nacional do Pico da Neblina to Novo Airão (see georeferenced localities in the S1 File).

*L. percarinatum* represents a complex of cryptic hybridizing lineages with different ploidy levels, some of which are parthenogenetic [64]. It was not possible to identify ploidy level for most individuals in this study, nor were observable differences attributable to ploidy for the few individuals with such information, or to the sampling site (See S1 File, data). Our results being consistent for the two currently accepted species, further studies can analyze finer effects of ploidy or geographic variation on our observed relationships, when ploidy data become more available.

**Experimental procedures.** We measured the lizard's VTmax three times ("trials") per individual: once within 48 h after capture, another time after dehydrating them for one night (8h), and once again after rehydrating them for another 24 h. Heretofore, we will refer to these trials as fresh, dehydrated, and rehydrated, respectively. After measuring the VTmax for the first time, in the morning, we dehydrated the lizards (range: 0.25–1.1 g in body mass) by keeping them overnight (from 23PM:1AM to 7–9:30 AM) within an individual, hermetic and opaque, plastic boxes (500ml), containing 400g of dry rice. We monitored the conditions within the boxes during the desiccation process. For that, we used two LogBox-RHT-LCD (NOVUS ©) with their probes inserted within two of these boxes, to record the temperature and relative humidity within the desiccating boxes every 30 minutes (total range: 22.9–27.8 C˚; 62.4–81.2%). The following morning, we measured their VTmax for the second time, lizards were brought back to their boxes, but now containing paper towels soaked with water. This procedure allowed lizards to rehydrate to hydration levels higher than when they were collected. Then, they had their VTmax measured again, by the morning of the third day of the procedure.

To avoid the assumption that freshly captured lizards, or rehydrated lizards, were at their fully hydrated state, we simply considered hydration level as the percentage of each individual's maximum mass obtained across the three measurements [5]. This simplification, combined with comparing each individual with itself along the three trials, allows testing whether higher hydration level, represented as a continuous variable, leads to exiting the heating chambers at hotter temperatures without actually knowing the fully hydrated state of each lizard. At each trial, lizards were weighed using a digital balance (precision 0.001 g), right before being heated in the heating chambers. Each day, the hydration/dehydration boxes were searched for defecation, and for the two cases where there were excrements, the individuals were discarded from the sample analyzed herein. The variation in body mass across the three trials is shown in Fig 1.

*Measurement of the voluntary thermal maximum (VTmax).* Lizards were heated following the protocol validated by Camacho et al., [27]. In brief, each individual was placed in a half-closed 400ml metallic can to warm it up. Before heating, lizards were left for a couple of minutes to assess whether they were willing to abandon the can or if they take it as a refuge. This procedure led to excerpt two individuals which were visibly unwilling to remain in it. Then, lizards were homogeneously heated by external flexwatt® tape wrapped around the sides and the bottom of the cans until the animals exited them. We registered the heating rate and starting temperature individually, allowing both parameters to vary independently (see the lack of correlation among them in S2 Table in S1 File). Later, we accounted for heating rate and starting temperature during the statistical analyses.

The VTmax was measured as the dorsal temperature, right over the shoulders of lizards, and just at the moment, lizards had that body part outside the can. Body temperature was

measured with the *pointer* tool of a C2 FLIR camera, set to 20 cm distance and 0.80 reflectivity. That setup gave below 0.6˚C error when comparing the temperature measured by the pointer and a T thermocouple pressed against the lizards' skin. This procedure was repeated for both species (see S1 File). Heterothermy (gradients in the temperature of different body parts) was evaluated by Camacho et al., for an identical heating container and similarly shaped and sized lizards, relatives of *Loxopholis* [69], and it can be disregarded. All procedures were approved by the ethics committee of the Universidade de São Paulo (protocol: 318/2018). All methods were carried out in accordance with relevant guidelines and regulations. the study was carried out in compliance with the ARRIVE guidelines. After this study, the animals were transferred to the responsible for another project on lizard genetics and taxonomy.

**Analyses.** *Effects of dehydration/hydration treatments on body mass.* We tested whether our treatments effectively altered the hydration state of our lizards. For that, we fitted a linear mixed model in which body mass was the continuous response variable, and trial was a categorical predictive factor with 3 levels: fresh, dehydrated, and hydrated. Individual identification and species entered as random effects. In this fashion, we only compared individuals of each species against themselves, accounting for interindividual variation in responses to the predictive factors [27, 71].

*Effect of dehydration and experimental parameters on the VTmax.* We used Akaike's information criterion (AIC) to select among six fitted mixed linear models [72]. In these models, the VTmax was sequentially predicted by hydration level and other factors: trial, starting temperature, body size, and heating rate (all of them continuous variables). We again used lizards' identification numbers as a grouping factor to account for repeated measures. In this way, each specimen works as a reference for itself across measurement trials. In addition, our analysis also accounts for factors that vary naturally in the field, such as the start temperature, heating rate, and body size. Aditionally, we evaluated if handling had any systematic effect on the VTmax by testing if trial order affected this trait (*i.e.*, whether lizards showed higher or lower VTmaxs as a result of repeating the same experiment). To address potential concerns with the statistical power of our analyses, we added a formal power analysis [73] for each model in the model selection procedure.

We performed all the analyses in R language (Vr. 4.0 R Core Team 2020). We used the *lmer* function from the lme4 package [74] for fitting the mixed models and do model selection; and the function *powersim* for power analysis (package simr [73]). Once a final model was selected, we used the function *lme* (package nlme [75]) that also provides p-values for the factors in the models. These p-values were calculated using the Kenward-Roger approximation that circumvents identified biases in Wald tests [76, 77].

We observed slight deviations from normality in the general model's residuals and for the separated model of *L. percarinatum*. In both cases, applying *rankit* transformation [78] to the response variable solved the issue (S3 Fig in S1 File). Since both approaches rendered identical conclusions, we kept the results for untransformed data in the manuscript for easiness of interpretation. Results under *rankit* transformation can be consulted at S2 Table in S1 File.

**Mapping potential effects of dehydration on the geography of perceived thermal risk across the Amazonian basin.** Herein we seek to simply illustrate how dehydration may alter geographic models of thermal risk. We generated these models based on the VTmax of hydrated and dehydrated lizards of *L. percarinatum* only, for which we had samples from multiple populations. We do not wish to provide a prediction of climatic vulnerability because we do not think reliable ones can yet be provided.

We used two approaches to make complementary maps of thermal risk. First, following methods in Recoder et al., we used a "simple mapping" approach to map the geographic extent and magnitude of warming tolerance margins [33] for all pixels of a raster file [45]. For

illustrative purposes, the warming tolerance (AKA thermal margins) was calculated as the median VTmax-Tenv. In this way, our thermal margins indicate the temperature difference between the estimate of the minimum available temperature at the hottest time of the year and the VTmax of half of a population of lizards, under the assumption of no geographic variation in VTmax. Here, Tenv is given by the bioclimatic variable bio 5 (10 years averaged, maximum yearly temperatures of air, measured in the shade). Bio 5 was acquired from the CHELSA database [79] (30-sec resolution), which accounts for topographic variation in temperature. Thus, to illustrate how changes in the VTmax may alter thermal risk, we assumed that populations would be more vulnerable the more Tenv exceeds the VTmax. In this approach, we created two maps, one with the median VTmax obtained from lizards over 95% of their fully hydrated mass and another for the VTmax of individuals dehydrated below 95% of their fully hydrated mass. We call these two arbitrary groups "hydrated" and "dehydrated" for illustrative purposes only.

Yet, interpolated air temperatures cannot describe the full thermal heterogeneity at the lizards' scale, and averages of the maximum temperatures do not show the full range of air thermal variation. In our second approach, Tenv represents the minimum attainable body temperatures, thus allowing us to account to some extent for the capabilities of lizards to thermoregulate and bury to avoid heating. We estimated the geographic extent (how many localities exhibit negative margins) and duration (how many hours per year negative margins are expected for each locality) of negative warming tolerance margins. For that, we used the Niche Mapper [37]. This algorithm uses heat and water transfer equations to translate climatic conditions into microclimatic conditions experienced by an animal of a certain shape, size and color, at different heights and soil depths, at a spatial resolution of centimeters [37, 80]. This approach thus allows a more accurate estimation of the body temperatures of small lizards protected underground in moist environments. In this study, we applied this algorithm for each raster cell of five-minute resolution, all across a shape of the Amazon basin provided by the WWF [81] (source: https://databasin.org/datasets/7c01a6d864fe4158b455c812ab040b1f/). Doing it for each 30 seconds cell of the CHELSA database would demand prohibitive calculation time. We also plotted major rivers to facilitate the spatial reference of locations. For that, we used freely available 1km resolution shapefiles, derived from Openstreetmaps (source: http://download.geofabrik.de).

To gauge the importance of changes in the VTmax for expected thermal risk, we compared their impact for geographic models of thermal risk with the expected impact of changes in other important environmental conditions. For that, we used the "hydrated" and "dehydrated" estimates of the VTmax, varying several parameters that the NicheMapR uses to calculate body temperatures. These variations were: a) body mass (0.525 g vs 0.628 g, for dehydrated versus hydrated lizards), the amount of surrounding surface that evaporates water (0 vs 100), the maximum depth to which the animal can bury (0 vs 10cm), the amount of precipitation (50% of current amount vs the current amount), and the maximum percentage in tree cover (80 vs 100%). We also fed the models with critical thermal maxima and preferred temperatures of *L. percarinatum* estimated by Diele-Viegas et al. [81]. Across all models, "virtual" lizards had access to at least a minimum shade typical from a Brazilian rainforest [82] (70%). For all the other parameters, we used the lizard's default in NicheMapR (see S8 Table in S1 File). Although NicheMapR can simulate the water budget of the lizards, we did not do that because it would require using data that we do not have, like the percentage of fully hydrated mass at which lizards become inactive. Since we did not wish to estimate if lizards would die from surpassing the critical thermal maximum, we did not alter the value of the critical thermal maximum and requested the model not to stop calculations if that value was exceeded. The resulting 64 maps are provided at the S1 Data "Maps". We acknowledge that changing these

factors with different magnitudes might result in different maps too. We simply try to show that dehydration-induced changes in the VTmax and related environmental parameters should affect the perceived geography of thermal risk.

We compared statistically the effects of these different parameters on three traits of thermal risk: The extent and duration of thermally stressful events, and the percentage of lizard populations affected by them, increased by several orders of magnitude as VTmax decreased, and were associated with dehydration more than any other parameter. For this, we fitted three separated generalized least squares models using the parameters as predictive factors of thermal risk. We used the gls function from the package nlme [83], in R language [51]. Scripts for all the tests and graphs shown herein are provided within the S9 and S11 Tables in S1 File.

## Supporting information

**S1 File. Data, supporting figures and tables, and scripts.**
(XLSX)

**S1 Data.**
(ZIP)

## Acknowledgments

We are grateful to Camila Moreira, Ivan Prates, José Cassimiro, José Mário Ghellere, Marco A. de Sena, Renato Recoder, and Sergio Marques-Souza for helping at the Rio Negro's expedition in 2018. Kátia Pellegrino, Camila Moreira e a Gabriela Farias for providing ploidy level data. Sergio Marques-Souza identified the species. Marco Antônio Marques de Souza assembled the solar energy system to allow experiments to be carried out on the boat. Richard Orton for proofreading the English language.

## Author Contributions

**Conceptualization:** Agustín Camacho.

**Data curation:** Agustín Camacho.

**Formal analysis:** Agustín Camacho.

**Funding acquisition:** Agustín Camacho, Tuliana O. Brunes, Miguel Trefaut Rodrigues.

**Investigation:** Agustín Camacho, Tuliana O. Brunes, Miguel Trefaut Rodrigues.

**Methodology:** Agustín Camacho, Tuliana O. Brunes, Miguel Trefaut Rodrigues.

**Project administration:** Agustín Camacho, Tuliana O. Brunes, Miguel Trefaut Rodrigues.

**Resources:** Agustín Camacho, Tuliana O. Brunes, Miguel Trefaut Rodrigues.

**Software:** Agustín Camacho.

**Supervision:** Miguel Trefaut Rodrigues.

**Validation:** Agustín Camacho.

**Visualization:** Agustín Camacho.

**Writing – original draft:** Agustín Camacho, Tuliana O. Brunes.

**Writing – review & editing:** Agustín Camacho, Tuliana O. Brunes, Miguel Trefaut Rodrigues.

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
