## [Decision Letter · Decision Letter 0]

20 Apr 2022

PONE-D-21-29853

Dehydration alters behavioral thermoregulation and the geography of climatic vulnerability in Amazonian lizards.

PLOS ONE

Dear Dr. Camacho,

Thank you for submitting your manuscript to PLOS ONE. After careful consideration, we feel that it has merit but does not fully meet PLOS ONE’s publication criteria as it currently stands. Therefore, we invite you to submit a revised version of the manuscript that addresses the points raised during the review process.

Please do take the time to give thoughtful consideration to the comments provided by the reviewers here. If substantial revisions are not made with respect to these comments, the manuscript will not be accepted. As you can see, one reviewer feels that previous reviews were not taken seriously.

We look forward to receiving your revised manuscript.

Kind regards,

Michael Sears, PhD

Academic Editor

PLOS ONE

Journal Requirements:

2. In your Methods section, please include a comment about the state of the animals following this research. Were they released, euthanized or housed for use in further research? If any animals were sacrificed by the authors, please include the method of euthanasia and describe any efforts that were undertaken to reduce animal suffering.

4. Thank you for stating the following in the Acknowledgments Section of your manuscript: "We are grateful to Camila Moreira, Ivan Prates, José Cassimiro, José Mário Ghellere, Marco A. de Sena, Renato Recoder, and Sergio Marques-Souza for helping at the Rio Negro’s expedition in 2018. Kátia Pellegrino, Camila Moreira e a Gabriela Farias for providing ploidy level data. Sergio Marques-Souza identified the species. Marco Antônio Marques de Souza assembled the solar energy system used to allow experiments to be carried out on the boat. This work was supported by the Fundação de Amparo à Pesquisa do Estado de São Paulo (FAPESP), processes 2016/03146-4 (TOB) and 2011/50146-6 (MTR); Conselho Nacional de Desenvolvimento Científico e Tecnológico (CNPq), process 301778/2015-9 (MTR); Coordenação de Aperfeiçoamento de Pessoal de Nível Superior/Programa Nacional de Pós Doutorado (CAPES/PNPD) fellowship #code: 0001 and Marie Curie Grant (897901) (ACG)."

"Agustín Camacho:CAPES/PNPD 001. MSCA:897901

Tuliana O. Brunes: FAPESP 2016/03146-4 

Miguel Trefaut Rodrigues: FAPESP: 2011/50146-6. CNPQ: 301778/2015-9.   

5. Please note that in order to use the direct billing option the corresponding author must be affiliated with the chosen institute. Please either amend your manuscript to change the affiliation or corresponding author, or email us at plosone@plos.org with a request to remove this option.

6. We note that Figure 3 in your submission contain map images which may be copyrighted. All PLOS content is published under the Creative Commons Attribution License (CC BY 4.0), which means that the manuscript, images, and Supporting Information files will be freely available online, and any third party is permitted to access, download, copy, distribute, and use these materials in any way, even commercially, with proper attribution. For these reasons, we cannot publish previously copyrighted maps or satellite images created using proprietary data, such as Google software (Google Maps, Street View, and Earth). For more information, see our copyright guidelines: http://journals.plos.org/plosone/s/licenses-and-copyright.

a. You may seek permission from the original copyright holder of Figure 3 to publish the content specifically under the CC BY 4.0 license.  

7. We note that Figure 1 in your submission contain copyrighted images. All PLOS content is published under the Creative Commons Attribution License (CC BY 4.0), which means that the manuscript, images, and Supporting Information files will be freely available online, and any third party is permitted to access, download, copy, distribute, and use these materials in any way, even commercially, with proper attribution. For more information, see our copyright guidelines: http://journals.plos.org/plosone/s/licenses-and-copyright.

Reviewers' comments:

Reviewer's Responses to Questions

**Comments to the Author**

1. Is the manuscript technically sound, and do the data support the conclusions?

Reviewer #1: Partly

Reviewer #2: Partly

2. Has the statistical analysis been performed appropriately and rigorously? 

Reviewer #1: Yes

Reviewer #2: Yes

3. Have the authors made all data underlying the findings in their manuscript fully available?

Reviewer #1: No

Reviewer #2: Yes

4. Is the manuscript presented in an intelligible fashion and written in standard English?

Reviewer #1: No

Reviewer #2: No

5. Review Comments to the Author

Reviewer #1: I already had the chance to review this manuscript when submitted to another journal, which unfortunately later rejected it. The aspect that strikes and saddens me the most is the number of comments I originally made that were wholly ignored. Indeed, most of my original comments had to be repeated in the present version. This means that the authors did not make any effort to improve their manuscript after the first rejection and merely resubmitted the manuscript somewhere else.

Hence, the authors disrespected a colleague who devoted various working hours to improve their manuscript. Now I discover that I wasted those hours since the authors ignored all of my suggestions. It is lousy and bad practice in science to ignore a reviewer's comments only because a journal rejected your manuscript. You could have submitted a much better version to PlosOne, instead of forcing me to waste even more time to repeat all my comments and force other reviewers to go through your original version. I sincerely hope that you will learn a lesson from this experience: there are not so many reviewers out there, after all, and your manuscript may end up in the hand of the same person again. Hence, revise your manuscript regardless of the final decision from the journal, even more so if it was rejected. Take advantage and acknowledge the amount of (free) work a reviewer devoted to trying to improve your work.

That said, and not willing to spend the same amount of time I invested on the first revision. The authors are not native speakers, and it shows. As for the first revision, I suggest many corrections (typically where the errors are painful to read), but I do not revise their English. Instead, I strongly advise an English revision by a native speaker, which has not been done. I'll also copy-paste all comments still valid.

Original major comments

As it stands, the manuscript needs a complete revision, and even after one is carried out, I believe it to be of more modest reach than the authors may feel it to be.

For starting, the English need a profound revision. Many grammar errors are present, and several sentences are hard to understand. I rarely recommend such a revision; however, the English quality is not good enough for publication. I pointed out some of the passages most clearly needing a change, but I encourage a more thorough revision, possibly done by a native English speaker.

Second, the sample size was small and unbalanced: one of the two species used was represented by less than ten individuals, while for the other, 19 individuals were tested. I can understand the difficulties in collecting data while working on a boat in the Amazonas. However, the limitations on statistical and biological reaching of a small sample size remain, no matter how hard it was to get the data. Hence, the reader must be cautioned about the inherent problems of small sample sizes, like the increased uncertainty of all estimates. Authors must account for this aspect, both in the result and the discussion.

Third, having a small sample size and a single population affect the mapping attempt. I applaud the effort to draw a map of overheating risk. Still, I have serious reserves that any significant conclusion could be drawn using just a few individuals from a single locality.

Specific comments (mostly repeated)

Abstract

L11: I would suggest using "hydro-thermoregulation" instead of "thermohydroregulate" throughout the manuscript. In my opinion, it would facilitate the reading. (Repeated)

L14: Replace "VTMAX". "VTmax".

L20: "which also reacted to start temperature and heating rates" is not clear. Please rephrase. (Repeated)

L21: remove the comma after "dehydration". (Repeated)

L23: remove the comma after "risk". (Repeated)

Introduction

L32: "thermophysiological" sounds weird. I suggest removing the word altogether since later on in the sentence, the authors explain the thermal dependence of performance. (Repeated)

L33: replace "Ex." with "e.g." throughout the text. (Repeated)

L34: I would replace "Dehydrated animals (ex. herps), exhibit" with "Dehydrated ectotherms may exhibit". (Repeated)

L35: "the high temperature that blocks locomotion": do you mean to say something like "exhibit lower critical thermal maxima, the limit which, if surpassed, would impede locomotion"? (Repeated)

L37: I would be more precise than "animals", especially if you already refer to ectotherms. (Repeated)

L40: the content of the brackets is odd: mammals is an order, chicken a family, and then two lizards species are called by their Latin names. Please rephrase it. (Repeated)

L42: "herp" is a colloquial term and should not be used in a scientific article. Please remove it from the text. (Repeated)

L47: "in" instead of "of". (Repeated)

L48: remove the comma after "both". (Repeated)

L49: place here the abbreviation VTmax. (Repeated)

L49: "a" instead of "one". (Repeated)

L56: "Might be overwhelmed" instead of "might have these limits overwhelmed". (Repeated)

L56: add "-" after "inter". (Repeated)

L61: add "in" before "the lizard". (Repeated)

L67: Add "By" before "combining". (Repeated)

L73: "Air temperature" instead of "The temperature of the air". (Repeated)

L72: I disagree with this statement. Most small body-sized ectotherms, like lizards, can easily escape into burrows, crevices and under rocks of various sizes to escape hot temperatures. Hence, air temperature in the shade is by no means the minimum temperature they have available. (Repeated)

L84: Provide references for the statement: "determines the availability of moisture and water for animals, potentially affecting dehydration rates, thermal tolerance, and the risk of exposure to extreme environmental temperatures".

Results

L104: "starting" instead of "start".

L104: rankit should be italicized throughout the text. (Repeated)

Discussion

L157: Add "-"after "intra". (Repeated)

L158: "of" instead of "in". (Repeated)

L159: This concept appears of the blue here for the first time. Briefly explain what do you mean by "anhomeostasis", instead of merely quoting a paper. This is the discussion section, after all. (Repeated)

L160: Remove the comma after "such as". (Repeated)

L171: "locomotion" instead of "speed"

L176: "research on" instead of "documentation of"

L179: "since active-thermoregulators are often found at them" is not clear. Please rephrase. (Repeated)

L183-184: The quote "20" does say the exact opposite of the authors. From the abstract of Sannolo et al. 2020, it can be read the dehydration lowered preferred body temperatures in all lizards tested. Please rephrase. (Repeated)

L184-185: the change in VTmax in the desert iguana is not due to dehydration since the water content did not change. Instead, after the injection, the Iguana suffered from electrolyte imbalance, which may influence the thermal parameters. Please rephrase. (Repeated)

L190: Discussing panting is out of topic, as it seems a clumsy attempt to increase auto-citation. Please delete this sentence. (Repeated)

L192: "Keeping the VTM up might help lizards (and other organisms)" is colloquial. Please rephrase. (Repeated)

L193: "maintaining": do you mean retaining?

L194: "keeping physiological performance high" is colloquial. Please rephrase. (Repeated)

L202-209: This paragraph on the differences between endotherms and ectotherms should be removed altogether. The differences between these two groups are so profound in any aspect considered (and run so deep in the phylogeny) that there is no point in comparing them to advocate for obvious differences in body size or water balance. (Repeated)

L210: This sentence is colloquial (define "high" and subject species), and no references are provided. Please rephrase and provide evidence or remove it. (Repeated)

L210: "in" instead of "at". (Repeated)

L212: "in" instead of "at". (Repeated)

L213: "inside a" instead of "at a"

L210-230 and 239-259: These whole paragraphs deal with specific details of the results. Hence, it should be moved at the very beginning of the discussion. The discussion section should flow from the details to the general debate, not the other way around.

Methods

L287: remove "as collected". (Repeated)

L301: "Brought back" instead of "taken back". (Repeated)

L302: Was this statement tested or just assumed? If tested, please provide numbers. (Repeated)

L305: Using the maximum weight is an assumption as well, since (unfortunately) there is no way to know for sure which weight represents the 100% hydrated condition. (Repeated)

L308: The term "turn" is confusing. I suggest using a more experimental wording, like "condition" or "trial". (Repeated)

L364: I do not see how body size is difficult to control experimentally in field studies. (Repeated)

344: Please italicize function names. (Repeated)

347: Please provide numbers for "slight deviations". (Repeated)

381: This first sentence of the paragraph si superfluous: state directly what you aim to achieve, not what you don't want to. (Repeated)

Reviewer #2: This manuscript reports on the effect of mild dehydration on the temperatures voluntarily tolerated by two species of Amazonian lizards. Based on those experimental results, the manuscript also reports output of two models of how those thermoregulatory responses to dehydration might affect risk of exposure to thermally challenging conditions across the geographic range.

As the authors state, the interactions between hydration and thermoregulation are an area that needs additional study. I think that this study provides some valuable data, and an interesting perspective, but I have some concerns about some assumptions, and comments about presentation.

The authors make a point in a couple of places (lines 53,396-7) that "exposure to it {VTMAX} can kill in a matter of hours", but that was for a species with VTMAX very high (42.5C) and close to CTmax (~43C), which is very different from the species in this study (VTMAX ~24-33C for both species; CTM>38C), which have a buffer of 5-8 degrees before reaching lethal/dangerous temperatures. This assumption that temps above VTmax are always dangerously high drives the results in the modeling analyses, and the conclusions that these species may be vulnerable under dryer and/or warmer conditions. Because of that large buffer, I'd like to see more justification that VTMax represents a dangerous threshold, or at least more discussion about its use as an illustration, rather than a realistic estimate of thermal vulnerability.

I was concerned about some assumptions in the modeling. In particular, the assumptions above, and the assumption that air temperature is the coolest temperature available, despite the models allowing for burrow use (soil temps can be cooler than air temps), and discussion in the manuscript about use evaporative cooling (panting, water loss from the surface, etc.). Because of this, it seems like the maps in Fig. 3 overstate the thermal risk of the species.

I found it hard to follow the justifications in the Intro and Discussion where temperate (or desert) species were used to justify ideas about low, wet tropical species. I think it is important to separate those more clearly in the text, because warm (sometimes arid) temperate locations create very different selective pressures than lowland wet tropics. In some cases, the manuscript uses references to mammals and birds to support ideas related to CTmax and VTmax, which are typically used to describe ectotherms.

The experimental data were good, but could be presented more effectively. Some results were relegated to supplemental figures (e.g. S1, Table S2). Lines 96-101, I wanted to see the percentage hydration levels to know how hydrated the animals were. It's more intuitively useful to see that they're 93% of maximum mass than knowing that they lost 0.075g when dehydrated. this made it very hard to know how dehydrated this setup made the animals. Hydration level also helps to understand the magnitude of the effect of dehydration on VTMax. It's given (line 110) as 0.12C per percent hydration, but nowhere was that extrapolated to the total change (e.g. from 93%-100% represents about 1C change in VTMax). This is especially important when the authors point to a 20% difference in the effect between species (line 218), which seems to be only a fraction of a degree.

Specific comments:

- define terms like: turn, start temperature, etc.

- Not sure what I was supposed to get from Table 1. Seems to be just a list of variables that effect the model output?

- The intro seemed to jumble CTmax and VTmax. Would help to clearly separate these concepts in the Intro

- Line 170-171; Pirtle et al. discusses behavioral alternatives to affect water loss, without thermoregulatory loss or energetic demand (Pirtle, Elia I., Christopher R. Tracy, and Michael Ray Kearney. "Hydroregulation: A neglected behavioral response of lizards to climate change?." Behavior of lizards. CRC Press, 2019. 343-374.)

lines 202-207: This paper is primarily relevant to ectotherms, so this paragraph is unnecessary and can be deleted

lines 231-237: I didn't see any discussion that justified these 6 characteristics. Please add justification for them.

lines 422-429: I think it's important to show that the CTM of this species is 6-10C above VTMax, and that the change in VTmax shown in the experiments is about 1C.

Check reference formatting.

Fig. 1. As much as I like seeing the study animal, this figure could be in the supplement.

Fig. 2. It took a long time to see that there was a grey line that was the overall response. Very hard to read this figure, and to understand what it is supposed to show. Consider presenting only the summary line (maybe with confidence intervals?) in the main body, and show the individual responses in the supplement.

Manuscript overall needs to be copy-edited by a native english speaker

6. PLOS authors have the option to publish the peer review history of their article (what does this mean?). If published, this will include your full peer review and any attached files.

Reviewer #1: No

Reviewer #2: No

---

## [Author Response · Author response to Decision Letter 0]

11 Aug 2022

PONE-D-21-29853

Dehydration alters behavioral thermoregulation and the geography of climatic vulnerability in Amazonian lizards.

Dear editor, please find below our answers to the reviewers, in bold.

Reviewers' comments:

Reviewer's Responses to Questions

Comments to the Author

1. Is the manuscript technically sound, and do the data support the conclusions?

Reviewer #1: Partly

Reviewer #2: Partly

2. Has the statistical analysis been performed appropriately and rigorously?

Reviewer #1: Yes

Reviewer #2: Yes

3. Have the authors made all data underlying the findings in their manuscript fully available?

Reviewer #1: No

Reviewer #2: Yes

4. Is the manuscript presented in an intelligible fashion and written in standard English?

Reviewer #1: No

Reviewer #2: No

 Reviewer #1: I already had the chance to review this manuscript when submitted to another journal, which unfortunately later rejected it. The aspect that strikes and saddens me the most is the number of comments I originally made that were wholly ignored. Indeed, most of my original comments had to be repeated in the present version. This means that the authors did not make any effort to improve their manuscript after the first rejection and merely resubmitted the manuscript somewhere else.

Hence, the authors disrespected a colleague who devoted various working hours to improve their manuscript. Now I discover that I wasted those hours since the authors ignored all of my suggestions. It is lousy and bad practice in science to ignore a reviewer's comments only because a journal rejected your manuscript. You could have submitted a much better version to PlosOne, instead of forcing me to waste even more time to repeat all my comments and force other reviewers to go through your original version. I sincerely hope that you will learn a lesson from this experience: there are not so many reviewers out there, after all, and your manuscript may end up in the hand of the same person again. Hence, revise your manuscript regardless of the final decision from the journal, even more so if it was rejected. Take advantage and acknowledge the amount of (free) work a reviewer devoted to trying to improve your work.

That said, and not willing to spend the same amount of time I invested on the first revision. The authors are not native speakers, and it shows. As for the first revision, I suggest many corrections (typically where the errors are painful to read), but I do not revise their English. Instead, I strongly advise an English revision by a native speaker, which has not been done. I'll also copy-paste all comments still valid.

Authors: Although we disagree with reviewer1’s main criticisms, which seem to derive from a misunderstanding of this work’s objective or methods, we followed his/her suggestions whenever we could. We did not want to disrespect anyone, but given that we oppose the main criticisms and the pressure for the mounting work, we decided to submit it to PlosOne, including only some changes that seemed most important. This time, we made large changes to the text to improve its clarity and avoid the same situation.

Original major comments

As it stands, the manuscript needs a complete revision, and even after one is carried out, I believe it to be of more modest reach than the authors may feel it to be.

For starting, the English need a profound revision. Many grammar errors are present, and several sentences are hard to understand. I rarely recommend such a revision; however, the English quality is not good enough for publication. I pointed out some of the passages most clearly needing a change, but I encourage a more thorough revision, possibly done by a native English speaker.

Authors: the text was previously reviewed by a non-native English teacher and this time by two native british speakers.

Second, the sample size was small and unbalanced: one of the two species used was represented by less than ten individuals, while for the other, 19 individuals were tested. I can understand the difficulties in collecting data while working on a boat in the Amazonas. However, the limitations on statistical and biological reaching of a small sample size remain, no matter how hard it was to get the data. Hence, the reader must be cautioned about the inherent problems of small sample sizes, like the increased uncertainty of all estimates. Authors must account for this aspect, both in the result and the discussion.

Authors: To account for the different challenges of small sample sizes during analysis, we conducted a rigorous test using individuals from two species to test the hypothesis: the VTmax is altered by hydration level in lizards, which is stated as the main objective of this manuscript. We used three repeated measures per individual, whose hydration level was experimentally manipulated. Our sample size is indeed lower for L. ferreira (9inds*3 trials =27) than for L. percarinatum (19*3=57), but we reached 80 observations to test our hypothesis and we are not comparing the two species. Instead, they enter as random grouping factor in our analysis (see methods).

Our version sent to PlosOne also included the following explanation in the methods: “This repeated-measures procedure has been validated to detect variations below 1ºC in the VTMAX of lizards with similar sample sizes[28]. Repeated-measures designs have increased statistical power that allow using relatively small samples[60] [28,60].” For the initial version sent to Plos one, we also included a formal analysis of statistical power for our samples, which further indicated that it reaches sufficient power level. This was made explicit in the methods and results: ex. “To avoid concerns with the statistical power of our analyses, we added a formal power analysis for each model in the model selection procedure. Power analysis identifies the probability of correctly rejecting the null hypothesis [62].”. Even though, we now also included the following disclaimer: “the sampling sizes could be considered somewhat low. However, given the changes in VTmax observed independently in almost every experimentally treated individual, in two different species, leaves us confident that the observed effects are real”.

Besides, our samples are indeed balanced (i.e. the compared treatments have identical sample sizes). We made explicit from the beginning that we are not comparing species in our tests, but the same individuals under different treatments. Thus, we also reject the criticism about unbalanced sampling.

Third, having a small sample size and a single population affect the mapping attempt. I applaud the effort to draw a map of overheating risk. Still, I have serious reserves that any significant conclusion could be drawn using just a few individuals from a single locality.

Authors: this is factually wrong. We declared the number of sites collected and their locations were provided in the supplementary material. We never did a map for L. ferreirai, the endemic species for which we have less sites.

“Thus, most L. ferreirai come from Igapós near to Boa Vista. Igapós are forests typical from Amazonian rivers, which experience seasonal floods. In turn, L. percarinatum were collected at Boa Vista, at terra firme forest, within the Parque da Neblina National Park, and at four sites in the west margins of the Rio Negro, across 500 km from the park to Novo Airão (see georeferenced localities in the supplementary file).”

Besides, we always stated that our maps were for L. percarinatum, the species from which we have collected data from several sites.

Specific comments (mostly repeated)

Abstract

L11: I would suggest using "hydro-thermoregulation" instead of "thermohydroregulate" throughout the manuscript. In my opinion, it would facilitate the reading. (Repeated)

Authors: Thermohydroregulation is a relatively new term, but already used in the literature. Searching for: “thermohydroregulation” gave about 39 results, some of which we cite.

Following the reviewer’s opinion, we made a search in Google scholar for hydro-thermoregulation. It rendered one result: [CITATION] Mechanisms~/Age# ig and De, elopment. 36 l 1986) 303-306 303 Elsevier Scientific Publishers Ireland Ltd.

Thus, we would wish to keep the option most acknowledged in the scientific literature and in the literature that we cite.

L14: Replace "VTMAX". "VTmax".

Authors: We exchanged VTMAX by VTmax. It is also called: VTM, Tvol, behavioral responses, and other terms in the scientific literature.

L20: "which also reacted to start temperature and heating rates" is not clear. Please rephrase. (Repeated)

Authors: we rephrased to:”and also changed with start temperature and heating rates”.

L21: remove the comma after "dehydration". (Repeated)

Authors: There was no such comma in the submitted version.

L23: remove the comma after "risk". (Repeated)

Authors: removed.

Introduction

L32: "thermophysiological" sounds weird. I suggest removing the word altogether since later on in the sentence, the authors explain the thermal dependence of performance. (Repeated)

Authors: searching for the term “thermophysiological” rendered 7.790 results in google scholar. Thus, we would wish to keep it, to remain in conformity with previous literature.

L33: replace "Ex." with "e.g." throughout the text. (Repeated)

Authors: Although we changed them, there was one left in the submitted version to PlosOne, that we now corrected.

L34: I would replace "Dehydrated animals (ex. herps), exhibit" with "Dehydrated ectotherms may exhibit". (Repeated)

Authors: We modified it to “Dehydrated amphibians and reptiles often exhibit..” 

L35: "the high temperature that blocks locomotion": do you mean to say something like "exhibit lower critical thermal maxima, the limit which, if surpassed, would impede locomotion"? (Repeated)

Authors: Yes, we intended that and changed the text to the following explanation for CTmax : “(i.e. a high body temperature which, if surpassed, would impede the individual’s locomotion [4–6])”

L37: I would be more precise than "animals", especially if you already refer to ectotherms. (Repeated)

Authors: We rephrased to: “Many ectothermic animals,”

L40: the content of the brackets is odd: mammals is an order, chicken a family, and then two lizards species are called by their Latin names. Please rephrase it. (Repeated)

Authors: We removed all the text about endotherms due to the criticisms of both reviewers.

L42: "herp" is a colloquial term and should not be used in a scientific article. Please remove it from the text. (Repeated)

Authors: we changed it to “amphibians and reptiles”.

L47: "in" instead of "of". (Repeated)

Authors: nor the two native speakers or the Grammarly app found any problem with of.

L48: remove the comma after "both". (Repeated)

Authors: removed.

L49: place here the abbreviation VTmax. (Repeated)

Authors: changed.

L49: "a" instead of "one". (Repeated)

Authors: changed.

L56: "Might be overwhelmed" instead of "might have these limits overwhelmed". (Repeated)

Authors: we left it as “Also, individuals not perfectly sheltered might be overwhelmed”.

L56: add "-" after "inter". (Repeated)

Authors: “interspecific” is correct according to the Collins dictionary.

L61: add "in" before "the lizard". (Repeated)

Authors: Added.

L67: Add "By" before "combining". (Repeated)

Authors: Added.

L73: "Air temperature" instead of "The temperature of the air". (Repeated)

Authors: Changed

L72: I disagree with this statement. Most small body-sized ectotherms, like lizards, can easily escape into burrows, crevices and under rocks of various sizes to escape hot temperatures. Hence, air temperature in the shade is by no means the minimum temperature they have available. (Repeated)

Authors: We precisely intended to account for thar in original manuscript: ”small ectotherms might still bury within moist microhabitats and experience lower maximum temperatures[32], making it important to evaluate how changes in thermal tolerance (Ex. in the VTMAX), and in the availability of such microhabitats, might alter our estimations of climatic vulnerability." 

L84: Provide references for the statement: "determines the availability of moisture and water for animals, potentially affecting dehydration rates, thermal tolerance, and the risk of exposure to extreme environmental temperatures".

Authors: we added the following reference: Davis, J. R., & DeNardo, D. F. (2010). Seasonal patterns of body condition, hydration state, and activity of Gila monsters (Heloderma suspectum) at a Sonoran Desert site. Journal of Herpetology, 44(1), 83-93.

Results

L104: "starting" instead of "start".

Authors: we changed that too.

L104: rankit should be italicized throughout the text. (Repeated)

Authors: changed.

Discussion

L157: Add "-"after "intra". (Repeated)

Authors: for the reason stated above, we did not adopt this suggestion.

L158: "of" instead of "in". (Repeated)

Authors: this sentence is no longer in the article.

L159: This concept appears of the blue here for the first time. Briefly explain what do you mean by "anhomeostasis", instead of merely quoting a paper. This is the discussion section, after all. (Repeated).

Authors: We exchanged anhomeostasis for its own definition and rephrased: “… except in the rare cases when individuals let their inner environment fluctuate largely [38].”

L160: Remove the comma after "such as". (Repeated)

Authors: removed

L171: "locomotion" instead of "speed"

Authors: changed

L176: "research on" instead of "documentation of"

Authors: changed

L179: "since active-thermoregulators are often found at them" is not clear. Please rephrase. (Repeated)

Authors: We removed this part

L183-184: The quote "20" does say the exact opposite of the authors. From the abstract of Sannolo et al. 2020, it can be read the dehydration lowered preferred body temperatures in all lizards tested. Please rephrase. (Repeated)

Authors: Sannolo et al are in reference 18, not 20. Their study show an effect on mean preferred temperatures, but the maximum temperatures accepted in the gradient, which are more comparable to a voluntary thermal maximum, remained at the same levels, something similar happens in le galliard’s study of Zootoca vivipara, now added to the manuscript, too. Thus, we rephrased to “Experimental dehydration did not lower the VTmax of the desert iguana [12], nor the maximum body temperatures of several temperate lacertidae species, studied in thermal gradients[15,18,21].”

L184-185: the change in VTmax in the desert iguana is not due to dehydration since the water content did not change. Instead, after the injection, the Iguana suffered from electrolyte imbalance, which may influence the thermal parameters. Please rephrase. (Repeated)

Authors: we rephrased to “However, desert iguanas did alter their VTmax in response to injections of concentrated saline solutions, leading to osmotic imbalance”

L190: Discussing panting is out of topic, as it seems a clumsy attempt to increase auto-citation. Please delete this sentence. (Repeated)

Authors: Panting is a behavior that combines thermal tolerance with water loss and is being used as a different behavioral measure of thermal tolerance. Thus, we believe it is appropriate here. Likewise, our review of the use of such trait across lizards is as relevant as early measurements of this parameter, which were also cited. We similarly acknowledged other reviews within topic (e.g., Tattersall`s, Pirtel’s, Curry-lindhal’s) throughout the manuscript.

L192: "Keeping the VTM up might help lizards (and other organisms)" is colloquial. Please rephrase. (Repeated)

Authors: we rephrased to “Maintaining a high VTmax might help lizards (and other organisms) behave more flexibly”.

L193: "maintaining": do you mean retaining?

Authors: we rephrased to “This could allow them to maintain feeding activity and higher metabolic rate, with the resultant ingestion of water in the food and enhanced metabolic water production”

L194: "keeping physiological performance high" is colloquial. Please rephrase. (Repeated)

Authors: this does not appear in our submitted manuscript to PlosOne.

L202-209: This paragraph on the differences between endotherms and ectotherms should be removed altogether. The differences between these two groups are so profound in any aspect considered (and run so deep in the phylogeny) that there is no point in comparing them to advocate for obvious differences in body size or water balance. (Repeated)

Authors: We deleted the reference to endotherms. 

L210: This sentence is colloquial (define "high" and subject species), and no references are provided. Please rephrase and provide evidence or remove it. (Repeated). 

Authors: we rephrased to: “Further, maintaining high body temperatures might be especially important for species living at places with short seasons for growth and reproduction (E.g. deserts, temperate regions, [38,42,49]”

L210: "in" instead of "at". (Repeated)

Authors: we rephrased to “Further, maintaining high body temperatures might be relevant for species living in places with short seasons for growth and reproduction”

L212: "in" instead of "at". (Repeated)

Authors: changed.

L213: "inside a" instead of "at a"

Authors: changed

L210-230 and 239-259: These whole paragraphs deal with specific details of the results. Hence, it should be moved at the very beginning of the discussion. The discussion section should flow from the details to the general debate, not the other way around.

Authors: We started the discussion with the main result, and then followed with secondary specific results. These lines 210-230 deal with implications of our experimental results for the understanding of effects of hydration on the VTmax. Thus, we would like to avoid starting by that part and keeping the structure as it is. The other reviewer did not see any problem with the structure of this manuscript’s discussion.

Methods

L287: remove "as collected". (Repeated)

Authors: removed

L301: "Brought back" instead of "taken back". (Repeated)

Authors: changed

L302: Was this statement tested or just assumed? If tested, please provide numbers. (Repeated)

Authors: As shown in that statement, it was observed that sometimes rehydrated lizards reached higher hydration levels, compared to recently captured ones. Thus, we did not test nor assumed a full hydration level. Using 100% for the most hydrated state of each individual allows testing for an effect of dehydration without knowing the fully hydrated state of each animal. Also, we added that: “To avoid the assumption that freshly captured lizards, or rehydrated lizards, were at their fully hydrated state, we simply considered their maximum weight obtained across the three measurements, as the fully hydrated weight. ”the numbers (weights) are provided in the supplementary material.

L305: Using the maximum weight is an assumption as well, since (unfortunately) there is no way to know for sure which weight represents the 100% hydrated condition. (Repeated)

Authors: we added: “This simplification, combined with comparing each individual with itself along the three turns, allows to identify whether higher hydration levels lead to exiting the heating chambers at hotter temperatures without actually knowing the fully hydrated state of each one.”

L308: The term "turn" is confusing. I suggest using a more experimental wording, like "condition" or "trial". (Repeated)

Authors: we changed turn by trial.

L364: I do not see how body size is difficult to control experimentally in field studies. (Repeated)

Authors: we did not argue that in our submitted manuscript to PlosOne. Instead we stated that it varies naturally and that we accounted for such natural variation in our analyses.

344: Please italicize function names. (Repeated)

Authors: italicized

347: Please provide numbers for "slight deviations". (Repeated)

Authors: provided

381: This first sentence of the paragraph is superfluous: state directly what you aim to achieve, not what you don't want to. (Repeated)

Authors: Despite this first sentence disclaimer, reviewer 2 was still concerned that our maps could be more than just illustrative. This makes us believe that this disclaimer is particularly relevant. We tried to find a suitable solution to Rev1 and Rev2 requests by taking the disclaimer to the end of the paragraph. 

Reviewer #2: This manuscript reports on the effect of mild dehydration on the temperatures voluntarily tolerated by two species of Amazonian lizards. Based on those experimental results, the manuscript also reports output of two models of how those thermoregulatory responses to dehydration might affect risk of exposure to thermally challenging conditions across the geographic range.

As the authors state, the interactions between hydration and thermoregulation are an area that needs additional study. I think that this study provides some valuable data, and an interesting perspective, but I have some concerns about some assumptions, and comments about presentation.

Authors: we thank the patience and time of the reviewer. We followed almost all of the suggestions and edited the text to ease his/her concerns.

The authors make a point in a couple of places (lines 53,396-7) that "exposure to it {VTMAX} can kill in a matter of hours", but that was for a species with VTMAX very high (42.5C) and close to CTmax (~43C), 

Authors: actually, apart from that one, other two lizard species, whose VTmax was 2 and 3 degrees below their respective CTmax also experienced deaths of individuals before 4h of exposure (see ref [28] in the manuscript). We explained this in the introduction [L70,80].

which is very different from the species in this study (VTMAX ~24-33C for both species; CTM>38C), which have a buffer of 5-8 degrees before reaching lethal/dangerous temperatures. 

This assumption that temps above VTmax are always dangerously high drives the results in the modeling analyses, and the conclusions that these species may be vulnerable under dryer and/or warmer conditions. Because of that large buffer, I'd like to see more justification that VTMax represents a dangerous threshold, or at least more discussion about its use as an illustration, rather than a realistic estimate of thermal vulnerability. 

Authors: Thanks, we did add more justification on that buffer distance and considerations for using the VTmax as a threshold [L 278-287]. Besides, we stated multiple times [104, 432, 447] across the text that our study and the maps are only illustrative.

I was concerned about some assumptions in the modeling. In particular, the assumptions above, and the assumption that air temperature is the coolest temperature available, despite the models allowing for burrow use (soil temps can be cooler than air temps), and discussion in the manuscript about use evaporative cooling (panting, water loss from the surface, etc.). Because of this, it seems like the maps in Fig. 3 overstate the thermal risk of the species.

Authors: we acknowledged the limitation of considering air temperature the minimum available [L 450-452] and actually applied a way to avoid the assumption that concerns reviewer2, which is using the Nichemapper, our second approach. The discussion goes about accounting for the animals ability to find a shelter and lower its maximum attainable temperatures [L292 and on]

I found it hard to follow the justifications in the Intro and Discussion where temperate (or desert) species were used to justify ideas about low, wet tropical species. I think it is important to separate those more clearly in the text, because warm (sometimes arid) temperate locations create very different selective pressures than lowland wet tropics. 

Authors: We need to compare our data from rainforest lizards with estimates of the VTmax performed before, those are from other species, mostly lizards from deserts and temperate regions. In the last version, we tried to leave the messages as clear as possible.

In some cases, the manuscript uses references to mammals and birds to support ideas related to CTmax and VTmax, which are typically used to describe ectotherms.

Authors: We excerpted all the references to mammals and birds.

The experimental data were good, but could be presented more effectively. Some results were relegated to supplemental figures (e.g. S1, Table S2). Lines 96-101, I wanted to see the percentage hydration levels to know how hydrated the animals were. It's more intuitively useful to see that they're 93% of maximum mass than knowing that they lost 0.075g when dehydrated. this made it very hard to know how dehydrated this setup made the animals.

Authors: Yes, because of space limitations, but we changed the picture of the animal by Fig S1. Readers can actually see VTMax directly related to hydration percentage in Figure 2. Yet, as we explain in the methods, there is no easy way to know the actual hydration state of each individual. That would require measurements of the blood of a baseline of fully hydrated lizards. What we do and is easy to do is to rank them according to their lost or gained body water. We explained that in the methods L[353-363]

Hydration level also helps to understand the magnitude of the effect of dehydration on VTMax. It's given (line 110) as 0.12C per percent hydration, but nowhere was that extrapolated to the total change (e.g. from 93%-100% represents about 1C change in VTMax). This is especially important when the authors point to a 20% difference in the effect between species (line 218), which seems to be only a fraction of a degree.

Authors: The total change in VTMax was actually much larger (2.5-3C on average) across the total change in hydration level studied (~77-100%). The largest changes reached 6 degrees. It is hard to tell what is more important for the animal, absolute or the relative changes. We thus presented both aspects (L126-145 and Fig 2). 

Specific comments:

- define terms like: turn, start temperature, etc.

Authors: Turn (now trial, attending the other reviewer) was originally defined in L406 and maintained in the new version. Starting temperature is now defined in L .60

- Not sure what I was supposed to get from Table 1. Seems to be just a list of variables that effect the model output?

Authors: This table does show the list of such parameters, but also the effects of changing these parameters on indices of thermal vulnerability. We explained that in the table’s legend and in the methods section L[486-489].

- The intro seemed to jumble CTmax and VTmax. Would help to clearly separate these concepts in the Intro

Authors: We did a few modifications. In sum, these measures are presented in two different paragraphs in the introduction section. We included amendments to try and make it clearer. L[34, 51]

- Line 170-171; Pirtle et al. discusses behavioral alternatives to affect water loss, without thermoregulatory loss or energetic demand (Pirtle, Elia I., Christopher R. Tracy, and Michael Ray Kearney. "Hydroregulation: A neglected behavioral response of lizards to climate change?." Behavior of lizards. CRC Press, 2019. 343-374.)

Authors: We added that reference and added this reference, too. 

Le Galliard, J. F., Rozen-Rechels, D., Lecomte, A., Demay, C., Dupoué, A., & Meylan, S. (2021). Short-term changes in air humidity and water availability weakly constrain thermoregulation in a dry-skinned ectotherm. Plos one, 16(2), e0247514.

slines 202-207: This paper is primarily relevant to ectotherms, so this paragraph is unnecessary and can be deleted.

Authors: we deleted that part.

lines 231-237: I didn't see any discussion that justified these 6 characteristics. Please add justification for them.

Authors: They are in there: Ex. trait 1: L221; trait 2: L695. Traits 3,4,5 and 6 arise from finding our species to have a VTmax highly reactive to dehydration, while the other studied lizards did not (Ex. L 225-230).

lines 422-429: I think it's important to show that the CTM of this species is 6-10C above VTMax, and that the change in VTmax shown in the experiments is about 1C.

Authors: We added that in L[762]. We do not know where is this 1C change coming from. We showed in Fig.2 that the average change is around 2.5 for L percarinatum and 3 for L. Ferreirai (look at model lines in figure 2).

Check reference formatting.

Fig. 1. As much as I like seeing the study animal, this figure could be in the supplement.

Authors: we changed it by our original Fig S1, which reviewer2 wanted to see in the manuscript.

Fig. 2. It took a long time to see that there was a grey line that was the overall response. Very hard to read this figure, and to understand what it is supposed to show. Consider presenting only the summary line (maybe with confidence intervals?) in the main body, and show the individual responses in the supplement.

Authors: In previous reviews of this manuscript, we have had different requests, including some who wanted the residuals and others that wanted the raw data, or the model line. To attend every perspective, we are showing both the raw data and the model expectations in the ms, and the residuals in the supporting material. We also showed the VTmax with respect to percentage hydration, which is precisely what reviewer 2 requested a few comments before this one. We tried to improve the figure and the legend, making the model line easier to see, and also plotted the confidence intervals based on the standard deviation, although they are so small that are hardly visible.

Manuscript overall needs to be copy-edited by a native english speaker

Authors: the text was copy-edited by two native speakers.

6. PLOS authors have the option to publish the peer review history of their article (what does this mean?). If published, this will include your full peer review and any attached files.

Do you want your identity to be public for this peer review? For information about this choice, including consent withdrawal, please see our Privacy Policy.

Reviewer #1: No

Reviewer #2: No

---

## [Decision Letter · Decision Letter 1]

2 Jan 2023

PONE-D-21-29853R1Dehydration alters behavioral thermoregulation and the geography of climatic vulnerability in Amazonian lizards.PLOS ONE

Dear Dr. Camacho,

Thank you for submitting your manuscript to PLOS ONE. After careful consideration, we feel that it has merit but does not fully meet PLOS ONE’s publication criteria as it currently stands. Therefore, we invite you to submit a revised version of the manuscript that addresses the points raised during the review process.

We look forward to receiving your revised manuscript.

Kind regards,

Daniel de Paiva Silva, Ph.D.

Academic Editor

PLOS ONE

Additional Editor Comments:

Dear Dr. Camacho,

After this new review round, both reviewers believe your manuscript may be accepted for publication in PLoS One after minor reviews. Still, as pointed out by both reviewers, there is a lot to improve in the text regarding the written English in your manuscript and general procedures, especially in the M&M section. Considering all that, I will provide a major review to your manuscript. Regarding the written English, I also suggest that you need to submit your text text to a third party company to review and improve your text in English.

After the improvements are taken care of, the manuscript will be submitted to a third review round, and hopefully, it will be accpeted by both reviewers.

Sincerely,

Daniel Silva

Reviewers' comments:

Reviewer's Responses to Questions

**Comments to the Author**

1. If the authors have adequately addressed your comments raised in a previous round of review and you feel that this manuscript is now acceptable for publication, you may indicate that here to bypass the “Comments to the Author” section, enter your conflict of interest statement in the “Confidential to Editor” section, and submit your "Accept" recommendation.

Reviewer #2: (No Response)

Reviewer #3: (No Response)

2. Is the manuscript technically sound, and do the data support the conclusions?

Reviewer #2: Yes

Reviewer #3: Partly

3. Has the statistical analysis been performed appropriately and rigorously? 

Reviewer #2: Yes

Reviewer #3: Yes

4. Have the authors made all data underlying the findings in their manuscript fully available?

Reviewer #2: Yes

Reviewer #3: Yes

5. Is the manuscript presented in an intelligible fashion and written in standard English?

Reviewer #2: No

Reviewer #3: Yes

6. Review Comments to the Author

Reviewer #2: General comments: This manuscript is much improved in some ways, and still needs some work in others. I found that most of my concerns about the content had been addressed, and therefore the science presented in the manuscript is much easier to evaluate, and seems to meet reasonable publication expectations. There are still a number of issues with language, some of which make it hard to understand what the authors are trying to say. These seem fixable, if the authors work more closely with someone experienced with scientific writing (especially in biology) in English (preferably a native speaker).

-Lit Cited needs copy editing for formatting (e.g. species names not italicized, strange capitalization in titles)

Line 20: “and also changed VTmax(?) Systematically”

Line 32: “despite it being”

Line 37: metabolic costs do what?

Line 39: “use behaviors that integrate temperature…”

Line 46: “associated with”

Line 47: hydration levels (instead of water levels)

Line 53: “temperature forces” and “seek cooler temperatures”

Line 56: no italics on Phrynosomatidae

Line 64: The Dupre paper was on desert iguanas, an active thermoregulator. Suggest stating the species name here, not referring to “desert lizards”, and pointing out that the species is not a thermoconformer. This sentence seems to fit better after the sentence on Hemiergis.

Line 70: Like I said in my previous review, the species referenced here has a CTM of 43.0-43.9°C and at VTmax of 42.5°C… 0.5°C below CTM! If you’re going to use this extreme case as your example to show that exceeding VTmax is dangerous, you need to be transparent and honest about how close those critical temperatures are to each other for this example species. Many other species, including many listed in the table in the Curry-Lindahl paper, have this difference in the range of 3-5 °C. Additionally, table 4 in that paper lists many examples of species that live for 24-48 hours (or more) when held at temperatures above VTmax. You don’t need to overstate your case that being above VTmax may be stressful to convince readers, so just be honest about this, rather than picking only the most extreme data. Or if you insist on using the most extreme, you need to make it obvious that you are referring to the most extreme case you could find.

Line 94: use “dampens” not damps

Line 101: Fig. 1 is a graph. Is there a figure missing?

Line 107: use “lizard weights”

Note: I reviewed the Methods section before Results and Discussion because the latter are impossible to understand without first reading the Methods

Line 310: replace period with comma

Line 311: “composed of leaf…”

Line 318: “nine adult L. ferreirai”

Line 319: Which species?

Line 328: “nor were observable differences attributable”

Line 330: “our results being”

Line 332: “data become”

Line 336: use “another” not more. Also add comma after dehydrated

Line 343, 344, 346, and many subsequent lines: I think recipients is supposed to be “chambers” or “boxes”? Recipients doesn’t make any sense.

Line 349: “allowed lizards to rehydrate to hydration levels”

I stopped copy-editing the English at this point (except where it affects meaning), but the rest of the Methods section has similar issues with wording and grammar that make it difficult to understand, and that do not meet expectations for clarity of writing in a manuscript acceptable for publication in PLOS ONE. I recommend again that the authors work more closely with someone experienced with scientific writing (especially in biology) in English, and preferably a native speaker.

Line 108: use “mass” instead of “weight” throughout

Line 109 and 110: use “difference from fresh mass”

Figure 2 legend (version under the figure): typo on “VTmaxax”, typo on “barrs”. Note – this figure is MUCH improved!

Line 127: Use “percentage” not percentual

Line 131: use “per degree/minute increase in heating rate”

Line 125 and 134: Both species had exactly the same range for VTmax? Both are given as 24.1 to 33.3C. This may be real, but is surprising, so I want to double check, especially because Fig. 2 suggests that VTmax may be higher in L. ferreirai than in L. percarinatum.

Figure 3 legend (version under the figure): I’m not sure what is meant by “the subtraction of the median VTmax”. Is this just the difference between yearly max and VTmax (yearly max – VTmax)?

Reviewer #3: Manuscript PONE-D-21-29853R1 reports on original research regarding the responses of two species of thermoconformers lizards to dehydration and environmental changes at the local scale and the interaction between these variables. The study presents a map of thermal risks for one of the species in the Amazon Basin, an output of an integrative approach between thermohydroregulation and models of the geography of thermal risks. Although innovative, it constitutes preliminary research on the topic for both species. My main concern is how far the authors go to generalize their findings. Nonetheless, the methods used are appropriate, and the results partially support the conclusions. Therefore, I recommend this for publication after the authors attend to the following minor commentaries.

General comments:

I recognize the efforts to improve sampling and that the work is not comparing both species. However, the low geographic extent of the data and the number of sites and species analyzed precludes any conclusions from being generalized to all Amazonian lizards or inferring about places of greater thermal risk throughout the Amazon Basin. Therefore, I strongly suggest a readjustment in the tone of writing so the manuscript is more precise regarding its objectives and the extent of the results presented.

Several processes described in the 'Methods' section are written confusingly and repetitively. I suggest reorganizing the subtitles and describing the steps in the chronological order in which they occurred to facilitate understanding. Below are some specific comments on these topics and others related to writing.

Specific comments:

Lines 1-2: To extrapolate the results to "Amazonian lizards", theoretically, it would be necessary to have at least a sample of species representative of the whole spectrum of general characteristics that define Amazonian lizards. As this is a research with two thermoconformers species, the extrapolation becomes inappropriate. I suggest identifying in the title that the research is about two species or populations of thermoconformers lizards.

Lines 4-6: Institution names in the native language (original names in Portuguese).

Lines 322-325: Please insert the number of collection sites for each species in the main text. Although the supplementary material has the coordinates of the sites, informing the number of sites will help the reader to visualize the extent of the data collected. For example, for L. percarinatum it was 5, correct? What about for L. ferreirai?

Line 333: The "Method" section begins with the subheading "Experimental procedures", which explains data collection in general. Next, the subtopics are presented: "Manipulation of lizards' hydration level" (line 340) and "Measurement of the voluntary thermal maximum" (line 363). It does not make much sense to separate the collection of dehydration information from VTMax, as both undergo the same treatments (fresh, dehydrated and rehydrated). As described, it is necessary to repeat the protocol information, such as the collection times. I suggest removing the cited subtopics and describing the processes all within "Experimental procedures" in the order in which both information is collected for each individual.

Lines 341-351: This passage is confusing. Example: on line 341, the authors state that the lizards were kept in plastic, opaque and hermetic boxes containing rice (for desiccation), and later in line 345, it says that the first measurement was obtained in the morning on the first day. Start the paragraph by describing the measurement on the first day, then proceed to the desiccation phase in the box and consecutive rehydration.

Lines 343-344: Why were temperature and humidity collected every half hour? The reason for this needs to be clarified as written.

Lines 369-370: Reallocate the number of variations in results.

Line 381: Replace “São Paulo University” with “Universidade de São Paulo”. The institution name is a proper noun and does not need to be translated into English.

Line 398 -400: Relocate Shapiro-Wilks test results to the "Results" section.

Lines 385-406: The "Analyses" section starts with a general description of the analyses and follows with subheadings explaining each statistical test in more detail. Line 386 of the "Analyses" section begins with a general description of each test. Later, on lines 403 to 404, it says, "For that, we fitted a linear mixed model in which body weight was the continuous response variable". This type of repetition of information leaves the text confusing and overwhelming. I suggest that the "Analyses" section start with a detailed description of each test (starting with the subtitles), and incorporate the information from lines 386 to 401. At the beginning of each subtitle, you can first explain the question to be answered, the variables used and the statistical test. Write in chronological order, aiming for simplicity, objectivity and avoiding confusion.

426 to 428: Is this transformation the same as in lines 398 to 401? If so, it is repetitive or confusing to mention it again. I suggest reorganizing the subheadings (as previously mentioned) and putting this information only once. As for the transformation, the mixed linear models are also presented more than once in the text.

Figure 3: In lines 431 to 435, it is specified that the maps in figure 3 are only illustrative (due to data limitations) of how much dehydration can change the geographic risk models and highlight the importance of dehydration in the process. However, the caption presents the first piece of information: "Maps predicting present thermal risk for Loxopholis percarinatum across the amazon basin", emphasizing not the process but the actual places of thermal risk. Therefore, I suggest rewriting the first sentence of the caption as something like: "Map of thermal risk variation, based on the response of lizards to dehydration".

Line 223: The cited reference (52) refers to a study about food items, microhabitat use, period of activity, reproductive cycle and fecundity of L. percarinatum. It does not address any issue related to thermoregulation.

7. PLOS authors have the option to publish the peer review history of their article (what does this mean?). If published, this will include your full peer review and any attached files.

Reviewer #2: No

Reviewer #3: No

---

## [Author Response · Author response to Decision Letter 1]

18 Jan 2023

Reviewer #2: General comments: This manuscript is much improved in some ways, and still needs some work in others. I found that most of my concerns about the content had been addressed, and therefore the science presented in the manuscript is much easier to evaluate and seems to meet reasonable publication expectations. There are still a number of issues with language, some of which make it hard to understand what the authors are trying to say. These seem fixable, if the authors work more closely with someone experienced with scientific writing (especially in biology) in English (preferably a native speaker).

-Lit Cited needs copy editing for formatting (e.g. species names not italicized, strange capitalization in titles)

A: references are now corrected.

Line 20: “and also changed VTmax(?) Systematically”

A: added.

Line 32: “despite it being”

A: changed

Line 37: metabolic costs do what?

A: Rise, clarified in text

Line 39: “use behaviors that integrate temperature…”

A: changed

Line 46: “associated with”

A: changed 

Line 47: hydration levels (instead of water levels)

A: changed

Line 53: “temperature forces” and “seek cooler temperatures”

A: changed the first but opted to keep retreat, as in the original citation and to avoid repetition.

Line 56: no italics on Phrynosomatidae

A: done

Line 64: The Dupre paper was on desert iguanas, an active thermoregulator. Suggest stating the species name here, not referring to “desert lizards”, and pointing out that the species is not a thermoconformer. This sentence seems to fit better after the sentence on Hemiergis.

A: Corrected. The study cited uses Sceloporus jarrovi and is not from Dupré. That sentence seems to be the conclusion of the paragraph, so we kept it in the end of the paragraph.

Line 70: Like I said in my previous review, the species referenced here has a CTM of 43.0-43.9°C and at VTmax of 42.5°C… 0.5°C below CTM! If you’re going to use this extreme case as your example to show that exceeding VTmax is dangerous, you need to be transparent and honest about how close those critical temperatures are to each other for this example species. Many other species, including many listed in the table in the Curry-Lindahl paper, have this difference in the range of 3-5 °C. Additionally, table 4 in that paper lists many examples of species that live for 24-48 hours (or more) when held at temperatures above VTmax. You don’t need to overstate your case that being above VTmax may be stressful to convince readers, so just be honest about this, rather than picking only the most extreme data. Or if you insist on using the most extreme, you need to make it obvious that you are referring to the most extreme case you could find.

A: The accusation of dishonesty here is unjustified. From the many examples pointed out by R2, we found only two unnoticed species that resisted over 24h in table 4 and are now added to the introduction text (L 86). Besides, according to the existing literature, the only extreme data here are these two species that the reviewer found. These two seem to be the exception, not the rule.

Besides, we had also already added that type of disclaimer at the discussion section.

“The VTmax offers higher sensitivity to detect thermal stress than the thermal limits: some desert lizards may show one to three degrees between the VTmax and the CTmax [27,31], but L. percarinatum may reach to 5 [40]. However, interspecific variation in the magnitude of this “thermal fear zone”, and the lack of published data on it, might be seen as a problem to compare thermal risk among species.”

And now added more discussion on this concern at L427.

“Thus, we tentatively suggest that whichever site at which the minimum available temperatures (I.e., at species’ shelters) overcome the VTmax for more than half day should induce enough stress to make a population vulnerable. This should still be true even in the case of species particularly well adapted to stressful heat (see introduction). Processes arising from failed thermoregulation at suboptimal refuges, and the concentration of individuals at thermal refuges, dehydration, and negative intra and interspecific interactions deriving from such concentrations could likely harm the stressed population under the proposed situation.”

Besides, CTmax is a far more geographically labile trait than VTmax, as data show in refs 28, 53, and in Clusella-Trullas, S., & Chown, S. L. (2014). Lizard thermal trait variation at multiple scales: a review. Journal of Comparative Physiology B, 184(1), 5-21. 

We do not know where Reviewer 2 obtained the CTmax data he/she is writing about but it seems more reasonable for a species from open habitats. The following reference cites most values under 40 degrees (even some under 30), although it also shows a couple of values over 50, which are likely typographic errors. Still, the average CTmax for Loxopholis percarinatum, including the likely typos, goes to 38 degrees. In the personal experience of the first author, who measured the VTmax of this genus in other situations, some Loxopholis (former Leposoma) died at temperatures below 38, during assays.

Diele-Viegas, L. M., Vitt, L. J., Sinervo, B., Colli, G. R., Werneck, F. P., Miles, D. B., ... & Ávila-Pires, T. C. (2018). Thermal physiology of amazonian lizards (reptilia: Squamata). PLoS One, 13(3), e0192834.

Line 94: use “dampens” not damps

A: changed

Line 101: Fig. 1 is a graph. Is there a figure missing?

A: Thanks. The figure (and now the call to it) had been removed, as requested by another reviewer.

Line 107: use “lizard weights”

A: changed

Note: I reviewed the Methods section before Results and Discussion because the latter are impossible to understand without first reading the Methods

A: that seems appropriate.

Line 310: replace period with comma

A: done

Line 311: “composed of leaf…”

A:Changed, “By” is also correct.

Line 318: “nine adult L. ferreirai”

A: s removed

Line 319: Which species?

A:The latter, now clarified

Line 328: “nor were observable differences attributable”

A: done

Line 330: “our results being”

A: done

Line 332: “data become”

A: done

Line 336: use “another” not more. Also add comma after dehydrated

A: done

Line 343, 344, 346, and many subsequent lines: I think recipients is supposed to be “chambers” or “boxes”? Recipients doesn’t make any sense.

A: done

Line 349: “allowed lizards to rehydrate to hydration levels”

A: done

I stopped copy-editing the English at this point (except where it affects meaning), but the rest of the Methods section has similar issues with wording and grammar that make it difficult to understand, and that do not meet expectations for clarity of writing in a manuscript acceptable for publication in PLOS ONE. I recommend again that the authors work more closely with someone experienced with scientific writing (especially in biology) in English, and preferably a native speaker.

A: We rechecked grammar and again copyedited the text with the help of an American English native.

Line 108: use “mass” instead of “weight” throughout

A: done

Line 109 and 110: use “difference from fresh mass”

A: done

Figure 2 legend (version under the figure): typo on “VTmaxax”, typo on “barrs”. Note – this figure is MUCH improved!

A: corrected

Line 127: Use “percentage” not percentual

A: Percentual is the correct adjective of unit.

Line 131: use “per degree/minute increase in heating rate”

Line 125 and 134: Both species had exactly the same range for VTmax? Both are given as 24.1 to 33.3C. This may be real, but is surprising, so I want to double check, especially because Fig. 2 suggests that VTmax may be higher in L. ferreirai than in L. percarinatum.

A:Thank you for pointing that, it is now corrected. The rest of the results are checked and ok.

Figure 3 legend (version under the figure): I’m not sure what is meant by “the subtraction of the median VTmax”. Is this just the difference between yearly max and VTmax (yearly max – VTmax)?

A: it now reads: “A and B represent the difference between the median VTmax of hydrated (A) or dehydrated (B) lizards and the mean of the yearly maximum air temperatures, at any pixel”

Reviewer #3: Manuscript PONE-D-21-29853R1 reports on original research regarding the responses of two species of thermoconformers lizards to dehydration and environmental changes at the local scale and the interaction between these variables. The study presents a map of thermal risks for one of the species in the Amazon Basin, an output of an integrative approach between thermohydroregulation and models of the geography of thermal risks. Although innovative, it constitutes preliminary research on the topic for both species. My main concern is how far the authors go to generalize their findings. Nonetheless, the methods used are appropriate, and the results partially support the conclusions. Therefore, I recommend this for publication after the authors attend to the following minor commentaries.

General comments:

I recognize the efforts to improve sampling and that the work is not comparing both species. However, the low geographic extent of the data and the number of sites and species analyzed precludes any conclusions from being generalized to all Amazonian lizards or inferring about places of greater thermal risk throughout the Amazon Basin. Therefore, I strongly suggest a readjustment in the tone of writing so the manuscript is more precise regarding its objectives and the extent of the results presented.

A: We thank the reviewer but we did not add new sites/individuals. Besides, in no section of the article we proposed to extrapolate the results to “all Amazonian lizards” as the reviewer states. The manuscript already included three paragraphs (in the objectives, in the methods, and at the conclusion) addressing the concern stated by reviewer 3.

In the objectives

“Later, we illustrated how changes in the VTmax, in the available soil depth for burying, in tree cover, and in precipitation, might alter the geography of perceived thermal risk for the more widely distributed L. percarinatum, across the Amazonian Basin.”

In the methods

“Herein we seek to simply illustrate how dehydration may alter geographic models of thermal risk. We generated these models based on the VTmax of hydrated and dehydrated lizards of L. percarinatum only, for which we had samples from multiple populations. We do not wish to provide a prediction of climatic vulnerability because we do not think reliable ones can yet be provided.”

At the conclusion

“In conclusion, data on patterns of species’ thermohydroregulation, aestivation, and field evaluations of presence in regions were models give opposing predictions, are badly needed before we can predict impacts of changes in the hydrothermal environment (e.g., through climate change, forest fires, or deforestation) on animal populations”

Several processes described in the 'Methods' section are written confusingly and repetitively. I suggest reorganizing the subtitles and describing the steps in the chronological order in which they occurred to facilitate understanding. Below are some specific comments on these topics and others related to writing.

Specific comments:

Lines 1-2: To extrapolate the results to "Amazonian lizards", theoretically, it would be necessary to have at least a sample of species representative of the whole spectrum of general characteristics that define Amazonian lizards. As this is a research with two thermoconformers species, the extrapolation becomes inappropriate. I suggest identifying in the title that the research is about two species or populations of thermoconformers lizards.

A: We changed the title to “in two Amazonian lizards”.

As we show, thermoconformer is a traditional but not quite valid term to characterize species, as species may behave as thermoconformers only while they are not thermally challenged.

Lines 4-6: Institution names in the native language (original names in Portuguese).

A: We will try to correct these in the electronic online form, also.

Lines 322-325: Please insert the number of collection sites for each species in the main text. Although the supplementary material has the coordinates of the sites, informing the number of sites will help the reader to visualize the extent of the data collected. For example, for L. percarinatum it was 5, correct? What about for L. ferreirai?

A: 4 and 2, added.

Line 333: The "Method" section begins with the subheading "Experimental procedures", which explains data collection in general. Next, the subtopics are presented: "Manipulation of lizards' hydration level" (line 340) and "Measurement of the voluntary thermal maximum" (line 363). It does not make much sense to separate the collection of dehydration information from VTMax, as both undergo the same treatments (fresh, dehydrated and rehydrated). As described, it is necessary to repeat the protocol information, such as the collection times. I suggest removing the cited subtopics and describing the processes all within "Experimental procedures" in the order in which both information is collected for each individual.

A: Ok, we combined the text.

Lines 341-351: This passage is confusing. Example: on line 341, the authors state that the lizards were kept in plastic, opaque and hermetic boxes containing rice (for desiccation), and later in line 345, it says that the first measurement was obtained in the morning on the first day. Start the paragraph by describing the measurement on the first day, then proceed to the desiccation phase in the box and consecutive rehydration.

A: done

Lines 343-344: Why were temperature and humidity collected every half hour? The reason for this needs to be clarified as written.

A: we do not have any important reason for using 30 min instead of 60 or 15. We just automatically monitored the conditions within the boxes and arbitrarily selected 30 min.

Lines 369-370: Reallocate the number of variations in results.

A: done

Line 381: Replace “São Paulo University” with “Universidade de São Paulo”. The institution name is a proper noun and does not need to be translated into English.

A: done

Line 398 -400: Relocate Shapiro-Wilks test results to the "Results" section.

A: Although these are “results”, they are not the results of the tests proposed in the objectives, but of a secondary test on premises performed along the process of the application of the analysis. To make the suggested insertion clear, we would need to also include methodological options in the results section and pollute that section with information that does not correspond to the objectives, nor affect the conclusions. Thus, we opted to keep the text like it was, none of the previous reviewers showed concerns with this point.

Lines 385-406: The "Analyses" section starts with a general description of the analyses and follows with subheadings explaining each statistical test in more detail. Line 386 of the "Analyses" section begins with a general description of each test. Later, on lines 403 to 404, it says, "For that, we fitted a linear mixed model in which body weight was the continuous response variable". This type of repetition of information leaves the text confusing and overwhelming. I suggest that the "Analyses" section start with a detailed description of each test (starting with the subtitles), and incorporate the information from lines 386 to 401. At the beginning of each subtitle, you can first explain the question to be answered, the variables used and the statistical test. Write in chronological order, aiming for simplicity, objectivity and avoiding confusion.

A: done

426 to 428: Is this transformation the same as in lines 398 to 401? If so, it is repetitive or confusing to mention it again. I suggest reorganizing the subheadings (as previously mentioned) and putting this information only once. As for the transformation, the mixed linear models are also presented more than once in the text.

A: the previous correction should solve this problem, too.

Figure 3: In lines 431 to 435, it is specified that the maps in figure 3 are only illustrative (due to data limitations) of how much dehydration can change the geographic risk models and highlight the importance of dehydration in the process. However, the caption presents the first piece of information: "Maps predicting present thermal risk for Loxopholis percarinatum across the amazon basin", emphasizing not the process but the actual places of thermal risk. Therefore, I suggest rewriting the first sentence of the caption as something like: "Map of thermal risk variation, based on the response of lizards to dehydration".

A. rewritten as suggested

Line 223: The cited reference (52) refers to a study about food items, microhabitat use, period of activity, reproductive cycle and fecundity of L. percarinatum. It does not address any issue related to thermoregulation.

A: at line 223 we have: 52. Crawford EC Jr. Brain and body temperatures in a panting lizard. Science. 1972;177: 431–632 433. So it seems to be correct.

---

## [Decision Letter · Decision Letter 2]

29 Mar 2023

PONE-D-21-29853R2Dehydration alters behavioral thermoregulation and the geography of climatic vulnerability in two Amazonian lizards.PLOS ONE

Dear Dr. Camacho,

Thank you for submitting your manuscript to PLOS ONE. After careful consideration, we feel that it has merit but does not fully meet PLOS ONE’s publication criteria as it currently stands. Therefore, we invite you to submit a revised version of the manuscript that addresses the points raised during the review process.

We look forward to receiving your revised manuscript.

Kind regards,

Daniel de Paiva Silva, Ph.D.

Academic Editor

PLOS ONE

Additional Editor Comments:

Dear Dr. Camacho,

After this second review round, one of the previous reviewers decided for the acceptance of your manuscript. Still, a new reviewer raised significant issues that prevent the publication of your manuscript at this time. Still, if you are able to correct and/or justify the issues raised by this second reviewer, both reviewers believe manuscript has a potential to be published in PLoS One after improvements are considered. Please resubmit your manuscript by May 13 2023 11:59PM or in an earlier date in case you are able to conduct the changes faster than I have previously imagined. Please do not hesitate to contact me in case you need further time or have any other doubt that needs to be taken care of.

Sincerely,

Daniel Silva.

Reviewers' comments:

Reviewer's Responses to Questions

**Comments to the Author**

1. If the authors have adequately addressed your comments raised in a previous round of review and you feel that this manuscript is now acceptable for publication, you may indicate that here to bypass the “Comments to the Author” section, enter your conflict of interest statement in the “Confidential to Editor” section, and submit your "Accept" recommendation.

Reviewer #3: All comments have been addressed

Reviewer #4: (No Response)

2. Is the manuscript technically sound, and do the data support the conclusions?

Reviewer #3: Yes

Reviewer #4: Yes

3. Has the statistical analysis been performed appropriately and rigorously? 

Reviewer #3: Yes

Reviewer #4: Yes

4. Have the authors made all data underlying the findings in their manuscript fully available?

Reviewer #3: Yes

Reviewer #4: Yes

5. Is the manuscript presented in an intelligible fashion and written in standard English?

Reviewer #3: Yes

Reviewer #4: Yes

6. Review Comments to the Author

Reviewer #3: (No Response)

Reviewer #4: This paper describes an experiment with two species leaf-dwelling, Amazonian lizard that were used in a laboratory-based study to examine how hydration-state influences voluntary maximum temperatures (VTmax). The authors used linear models to compare between hydrated, dehydrated, and rehydrated lizards to see how hydric state influence VTmax. The authors also included important factors such as heating rate and starting temperature to create their models. Additionally, they used the NicheMap package to make predictions about population risks depending on rainfall and temperature patterns in the Amazon. The manuscript may prove important to thermal biologists looking to include other important physiological variables (ie hydration) when predicting species responses to fluctuating thermal environments. The flow of the manuscript and reporting/presentation of the data need some improvement and I have made some specific suggested changes below as well as some general comments/concerns about major portions of the manuscript. Overall, I found this to be a compelling ms. However, my major concerns are with the logical flow of the introduction, your definition of hydrated versus dehydrated, and using the discussion so that readers understand the relevance of your results as a means to validate previous work or propose new frameworks. I commend the authors on already going through several rounds of review and I hope that they agree my suggested changes are feasible and improve the manuscript.

General comments:

Introduction:

-Earlier in lines L33-34 it seemed as though this paper was focusing on physiological responses to thermal and hydric environmental conditions. Most of your intro reads as such too. Then around ~L92 the paper switches to focus on using environmental moisture as a factor to further explore thermal traits using a modeling approach. Then the final sentence switches back to focusing on hydration. Environmental moisture and available water to maintain hydric state (dietary water, drinking water, and in some cases metabolic water) are very different things and it is confusing to suddenly go back and forth. If you decide to keep this paragraph in the intro about ‘environmental water’ it should be physiologically-based.

-Furthermore, your experiment uses categorical variables of hydric state (which are problematic, see general comment below in the results section) as factors to explain thermal conditions for lizards. No part of your experiment details hydro-regulation in these animals and I suggest removing most of the references to hydroregulation and instead frame your experiment as one where you consider the role hydration plays in animals abilities to acclimate, or not, to varying thermal constraints. This is a paper about factors that explain VTmax, as the title suggests, but starting in the abstract (L12) you introduce thermo-hydroregulation which is very different and involves organisms maintain homeostasis in regards to hydration and Tb (both separately and combined).

Results

-Please be consistent with significant figures throughout the ms

-It is unclear but appears as though hydrated is 95.0-100% HL and dehydrated is 0-94.9% HL. There is also never any definitive of HL. I assume that is hydrated level. And is it based on mass entirely? There is a definition of hydration level in L364-366 but it is not made clear Furthermore, This arbitrary grouping is problematic as there is no justification for why an animal is ‘hydrated’ at 95.0% but suddenly dehydrated when it reaches 94.9%. In this sense, that 0.1% difference has a major impact around 95% but means nothing when comparing 96.2% to 96.3%. Consider having a larger difference here of at least 1-2%. And is there any justification for why only a 5% decrease in body mass means that animals are dehydrated? Clinical levels of dehydration are typically only after 15-20% body mass loss. I’m glad to see the authors did not push the animals that much, but you need more in the text for why these animals are considered ‘dehydrated’.

-It is unclear why Table 1 is included in the ms. It is only referenced once in the results along with separate suppl. Tables. Since it doesn’t appear to be important to the overall story (ie it isn’t mentioned in the discussion) consider moving it as a supplementary table.

Results/Discussion

-Similarly, Figures and 1 and 2 are only mentioned once each in your results. The discussion is an important chance for you to put your results in the context of previous work and allow you the change to justify any of your claims with evidence. Yet, the only figure or table you mention is figure 3 twice (L255 and L295). If you keep table 1, and figs 1-2 in the manuscript they need to be included in the discussion so that readers understand the relevance of your results to validate previous work or propose new frameworks (as you do in L273-277). As written, I see no reason to have anything except figure 3 in the final manuscript.

Discussion

- L199-202 and throughout the discussion- It is unclear why this is being presented as one-sided. As written, you suggest that animals become dehydrated which lowers their VTmax. It could just as easily be explained as animals choose as higher VTmax but it comes at the cost of dehydration. Both potential scenarios need to be considered, especially considering your metric for dehydration (~%5 loss in body mass) might not signify an ecologically relevant level of water loss. For example, a normosomic animal might behaviorally select a higher temperature to increase metabolism but at the cost of increased water loss. These animals then may retreat to cooler areas to avoid dangerous levels of water loss or even to rehydrate so they can return to warmer locations. The discussion is written very thermal-centric, which is fine, but you should at least consider this entire story from an equally hydric lens.

Specific Comments:

L34- Please provide reference(s) for physiological performance being influenced by hydration. I assume the physiological performance being influenced by temperature is specific to ectotherms but that is not clear in this sentence. It is also unclear if you mean both temperature and hydration (as the example you give suggests) or separately and combined. Please clarify.

L35- I should be capitalized and i.e. should be in italics as it is latin

L37-39: reword this sentence. Metabolic rates can definitely increase but EWL isn’t necessarily as closely related in squamates, or even across ectotherms. This might be specific to tropical forest species, if so this needs to be stated clearly.

L39: remove this generalization with ecotherms because it simply isn’t true. There are numerous studies demonstrating species resilience when faced with higher temperatures and dehydration.

L41- remove ‘such’ from this sentence

L43- this portion is confusing, please reword and clarify what you mean by animals “needing” higher temperatures. Do you mean that they don’t start to pant or urinate until temperatures are higher than normal because they’re dehydrated?

L45- It is confusing to suddenly have an example with just thermoregulation in between sentences with thermohydroregulation. Please re-order.

L50- consider re-writing so that it reads “…water stress could help us understand how climatic restrictions may influence their activity and distribution.”

L52- the previous sentence stresses the importance of considering both hydric and thermal constraints. But then this sentence emphasizes just a thermal trait measurement. Re-word this sentence to avoid confusion.

L62- it is not clear, as written, the “them” that these typical thermoconformers are responding to. Oxygen conentrations? Typical thermal assays? Please clarify

L64- change “would” to ‘could’

L66- The saline injections needs to be clarified or removed. If you keep it, you need to define what you mean earlier in the introduction by dehydration. Often times it is simply referring to hyperosmolality unless you want to define it as hypernatremia, as is suggested by this reference/sentence.

L68- consider removing “magnitude and” to make this sentence crystal clear.

L69-71- this just means that some species VTmax is at or very close to their CTmax

L72-76- you need to at least address species specific VTmax versus CTmax. It’s exclusion here is confusing. Especially considering the next paragraph starts with thermal safety margins and a mention of CTmax. I understand not taking up too much space in this paper for brevity but your justification (L69-71) for species living in potentially dangerously warm environments seems like cherry picking.

L86- Please define or explain what you mean by ‘dry-skinned ectotherms’ I assume snakes and lizards

L87- change to “might bury themselves within ”

L129-130 AND L138-139: I would prefer if you included descriptive stats here. However, they can also be found in Table S2, which is okay. If you don’t include descriptive statistics here, please at least include something like “(all p<0.05)” since you use ‘significantly affected by’

L132 and throughout- include units for the mean body mass, I assume 0.694 g. (also on L134 an a few other places in the results).

L151- what are negative thermal margins? I can’t find in the methods what this means, please explain somewhere in the manuscript.

L168-170- as written this is confusing, consider re-writing so that it reads “The extent and duration of thermally stressful events, and the percentage of lizard populations affected by them, increased by several orders of magnitude as VTmax decreased, and were associated with dehydration more than any other parameter.”

L174- change to “had a larger affect”

L176- change to ‘changes in mass associated with dehydration”

L176- I assume this is in reference to hydration in animal who are 10cm deep and not dehydration in general? Please make sure this is clear here.

L196- what is meant by internal environment?

L207- what is the “receptively” in reference to? If the citations, then it needs to be within the brackets.

L209-219- unless you have tropical/temperature lizard-specific references here, remove metabolic water as that is typically negligible in reptiles except in some cool examples of desert herbivores/insectivores.

L216-240- It is unclear what this paragraph adds to the discussion. This paragraph details studies in other species (large, long-lived desert lizards) that are very different from the two tropical species used in your experiment (small, short-lived leaf litter lizards). Consider removing it entirely for brevity.

L226- Please see work by Hazard (e.g., 2001, PBZ) to update these sentences. Desert iguanas have salt glands which is not relevant to the two species your experiment used.

L245-246- osmotic pressure is measured in the plasma portion of blood. Similar to my comment on L66, it is unclear here the difference between hyperosmolality and ‘dehydrating processes’

L256- what is hydration rate? That is not mentioned anywhere in the manuscript and it is unclear what you are referencing. How fast they can absorb water in the GI tract? Water loss rates? Please clarify

L273- You did not measure plasma osmolality so this statement is not justified

L274- the only known lizard with a water reservoir is the Gila Monster who can use their urinary bladder, similar to desert tortoises. Having a greater total amount of body water (because of being large bodied) does not equal a meaningful amount of great total body water as this value is proportional and typically 65-75% across terrestrial vertebrate taxa.

L285- if you use quotations, please add a citation directly after so readers know who you are quoting here.

L322- change to “…who live over and within…”

L354- is this absolute humidity or relative humidity?

L374- it is unclear what you’re trying to say here, all individuals, except two, stayed in the can? Please re-word for clarity.

L375- change to “…tape wrapped around the side…”

L407- change “starting temperature, heating rate, and body size.”

L408- change “Besides” to ‘additionally’

L410- change “a reviewer’s” to ‘potential’

7. PLOS authors have the option to publish the peer review history of their article (what does this mean?). If published, this will include your full peer review and any attached files.

Reviewer #3: No

Reviewer #4: **Yes: **George Brusch IV

---

## [Author Response · Author response to Decision Letter 2]

3 May 2023

Dear Dr. Camacho,

After this second review round, one of the previous reviewers decided for the acceptance of your manuscript. Still, a new reviewer raised significant issues that prevent the publication of your manuscript at this time. Still, if you are able to correct and/or justify the issues raised by this second reviewer, both reviewers believe manuscript has a potential to be published in PLoS One after improvements are considered. Please resubmit your manuscript by May 13 2023 11:59PM or in an earlier date in case you are able to conduct the changes faster than I have previously imagined. Please do not hesitate to contact me in case you need further time or have any other doubt that needs to be taken care of.

Sincerely,

Daniel Silva.

Dear Daniel, 

thanks for the chance to resubmit. Since reviewer number three has already expressed satisfaction, we move on to discuss the concerns voiced by reviewer number four.

\\

Águs Camacho on behalf all authors

Reviewer #4: This paper describes an experiment with two species leaf-dwelling, Amazonian lizard that were used in a laboratory-based study to examine how hydration-state influences voluntary maximum temperatures (VTmax). The authors used linear models to compare between hydrated, dehydrated, and rehydrated lizards to see how hydric state influence VTmax. The authors also included important factors such as heating rate and starting temperature to create their models. Additionally, they used the NicheMap package to make predictions about population risks depending on rainfall and temperature patterns in the Amazon. The manuscript may prove important to thermal biologists looking to include other important physiological variables (ie hydration) when predicting species responses to fluctuating thermal environments. The flow of the manuscript and reporting/presentation of the data need some improvement and I have made some specific suggested changes below as well as some general comments/concerns about major portions of the manuscript. Overall, I found this to be a compelling ms. However, my major concerns are with the logical flow of the introduction, your definition of hydrated versus dehydrated, and using the discussion so that readers understand the relevance of your results as a means to validate previous work or propose new frameworks. I commend the authors on already going through several rounds of review and I hope that they agree my suggested changes are feasible and improve the manuscript.

We appreciate Dr. Brusch's review. To support earlier research or suggest new frameworks, we revised the introduction's thought flow, highlighted the distinction between hydrated and dehydrated animals, and tried to strengthen the discussion part. To learn more about how we achieved it, please read our answers to his remarks.

General comments:

Introduction:

-Earlier in lines L33-34 it seemed as though this paper was focusing on physiological responses to thermal and hydric environmental conditions. Most of your intro reads as such too. Then around ~L92 the paper switches to focus on using environmental moisture as a factor to further explore thermal traits using a modeling approach. Then the final sentence switches back to focusing on hydration. Environmental moisture and available water to maintain hydric state (dietary water, drinking water, and in some cases metabolic water) are very different things and it is confusing to suddenly go back and forth. If you decide to keep this paragraph in the intro about ‘environmental water’ it should be physiologically-based.

The introduction cannot be solely physiological because we are linking changes in physiological traits (VTmax and dehydration) to changes in ecological traits (vulnerability) in this study.

As a matter of fact, we start the introduction by writing about protocols to evaluate thermal vulnerability, for which thermohydroregulation data are needed. We must write about environmental water because it is a major source of hydration for lizards and other animals. Thus, we need this paragraph to introduce the modeling approach and the included variables in this study, which is why it does not look physiologically based, but rather “eco”. The paragraph in question is the last one right before the objectives, which obligately represents both the physiological and ecological variables of the study. That paragraph is precisely there to avoid excessive switching between topics along the introduction. Nonetheless, although we cannot prescind from the paragraph, we tried to give it a more physiological perspective. Hopefully, the reviewer finds the changes satisfactory.

-Furthermore, your experiment uses categorical variables of hydric state (which are problematic, see general comment below in the results section) as factors to explain thermal conditions for lizards. No part of your experiment details hydro-regulation in these animals and I suggest removing most of the references to hydroregulation and instead frame your experiment as one where you consider the role hydration plays in animals abilities to acclimate, or not, to varying thermal constraints. This is a paper about factors that explain VTmax, as the title suggests, but starting in the abstract (L12) you introduce thermo-hydroregulation which is very different and involves organisms maintain homeostasis in regards to hydration and Tb (both separately and combined).

Our categorical predictor represents three different hydration treatments, not hydric state: fresh (measured as collected), dehydrated, indicating that the animal underwent a dehydration treatment for one night, and rehydrated, indicating that the animal underwent a 24-hour rehydration period. We used a continuous variable, hydration level, as the percentage of the maximum weight measured for the animal across the three treatments. We do not see problems with this approach. Nonetheless, we made an effort to be more clear about this.

According to the definition of thermohydroregulation, by Rozen-Rechels, this term represents the interaction of hydration level and temperature in behavior. Since we are measuring the voluntary thermal maximum of each individual three times at different hydration levels, we consider we are observing thermohydroregulation. We are not observing acclimation as Dr. Brusch suggests since we are not keeping individuals under acclimation treatments, in which the state of the animal typically remains the same (i.e., same temperature) and different at each treatment level. Our treatments (fresh, dehydrated, and rehydrated) are dehydrating/rehydrating animals. Then, hydration level is used as an explanatory variable in the model to predict to predict induced changes in behavioral thermoregulation (Thus thermohydroregulation)

We agree that this paper shows factors that make the VTMax vary and how that variation translates into geographic predictions of thermal vulnerability.

Rozen-Rechels 2019 defines thermohydroregulation as integrating water levels and temperature in behavior. Since we are observing changes in the maximum thermal levels voluntarily tolerated, according to water levels, we (and reviewer 3, since he/she approved this ms for publication) consider that the observed behavior can correctly be termed thermohydroregulation.

Results

-Please be consistent with significant figures throughout the ms

Sorry, we are not sure what "significant figures" mean. We double-checked the text's usage in the names of the figures for the main figures (Fig) and the supplementary figures (Fig S). 

-It is unclear but appears as though hydrated is 95.0-100% HL and dehydrated is 0-94.9% HL. There is also never any definitive of HL. I assume that is hydrated level. And is it based on mass entirely? There is a definition of hydration level in L364-366 but it is not made clear.

We changed HL to hydration level in all instances. We also clarified the explanation of HL to: “we simply considered hydration level as the percentage of each individual’s maximum mass obtained across the three measurements” (L 364-370). Thus, hydration level, used in the model predicting the VTMax, is a continuous variable, not categorical nor a group.

Furthermore, This arbitrary grouping is problematic as there is no justification for why an animal is ‘hydrated’ at 95.0% but suddenly dehydrated when it reaches 94.9%. In this sense, that 0.1% difference has a major impact around 95% but means nothing when comparing 96.2% to 96.3%. Consider having a larger difference here of at least 1-2%. And is there any justification for why only a 5% decrease in body mass means that animals are dehydrated? Clinical levels of dehydration are typically only after 15-20% body mass loss. I’m glad to see the authors did not push the animals that much, but you need more in the text for why these animals are considered ‘dehydrated’.

We did not compare animals at 96.2% to 96.3%. During the experiments, the grouping (fresh, dehydrated and hydrated) represents which of the three treatments defined in lines 509-511 had each individual been subjected to. Because these treatments altered the hydration state of each individual differently, we could later use hydration level as a continuous variable predicting the VTmax.

We only splitted animals into two hydration groups for mapping thermal vulnerability. The hydrated category takes the median weight and VTmax of animals with hydration level above 95%. The dehydrated category uses mean weight and VTmax of animals hydrated below 95% (i.e., from there down to the minimum hydration level achieved across our animals). Thus, again, we are not comparing animals at 96.2% to 96.3%, nor at 95% with 94%. As written:

“For this approach, we created two maps, one using the median mass and VTmax obtained from lizards over 95% of their fully hydrated mass, and another for the weight and VTmax of individuals dehydrated below the 95% of their fully hydrated mass. We call these two arbitrary groups “hydrated” and “dehydrated” for illustrative purposes only.”

-It is unclear why Table 1 is included in the ms. It is only referenced once in the results along with separate suppl. Tables. Since it doesn’t appear to be important to the overall story (ie it isn’t mentioned in the discussion) consider moving it as a supplementary table.

A prior referee requested the inclusion of this table. Because there is a margin and the third referee agrees. We'd rather leave it at that. In addition, direct references to the results' figures and tables in the discussion section are frequently discouraged.

Results/Discussion

-Similarly, Figures and 1 and 2 are only mentioned once each in your results. The discussion is an important chance for you to put your results in the context of previous work and allow you the change to justify any of your claims with evidence. Yet, the only figure or table you mention is figure 3 twice (L255 and L295). If you keep table 1, and figs 1-2 in the manuscript they need to be included in the discussion so that readers understand the relevance of your results to validate previous work or propose new frameworks (as you do in L273-277). As written, I see no reason to have anything except figure 3 in the final manuscript.

Figures 1-2 show the efficiency of the treatments on hydration levels as well as the influence of the main factors tested experimentally. As a result, including them in the results section appears to be necessary to display to the readers the evidence that is available. Regarding referring figures in the discussion part, we followed tutorials that recommended the contrary

https://www.sfedit.net/how-to-write-a-discussion-section-for-a-scientific-paper/#:~:text=The%20discussion%20section%20is%20not,presented%20in%20the%20results%20section.

We nonetheless recognize there is a diversity of opinions on the matter <https://academia.stackexchange.com/questions/42032/referencing-figures-or-tables-in-discussion-section> being the one we took potentially the dominant one: reference to figures and tables should be made in the results section only as in Plos One’s manuscripts <https://journals.plos.org/plosone/article?id=10.1371/journal.pone.0271076 & https://journals.plos.org/plosone/article?id=10.1371/journal.pone.0261173>.

Given the fact that ref 3 has already approved the ms, we would wish to avoid incorporating changes that may result in new objections.

Discussion

- L199-202 and throughout the discussion- It is unclear why this is being presented as one-sided. As written, you suggest that animals become dehydrated which lowers their VTmax. It could just as easily be explained as animals choose as higher VTmax but it comes at the cost of dehydration. Both potential scenarios need to be considered, especially considering your metric for dehydration (~%5 loss in body mass) might not signify an ecologically relevant level of water loss. For example, a normosomic animal might behaviorally select a higher temperature to increase metabolism but at the cost of increased water loss. These animals then may retreat to cooler areas to avoid dangerous levels of water loss or even to rehydrate so they can return to warmer locations. The discussion is written very thermal-centric, which is fine, but you should at least consider this entire story from an equally hydric lens.

Since Dr. Brusch is referring here to different items, we respond them one by one:

As written, you suggest that animals become dehydrated which lowers their VTmax. It could just as easily be explained as animals choose as higher VTmax but it comes at the cost of dehydration.

Yes, we suggest that. The results strongly indicate for both species that VTmax increases as hydration level does (See figure 2 and Table 1). In turn, we did not evaluate the dehydration costs at different temperatures, so we cannot say that animals choose as higher VTmax but it comes at the cost of dehydration. 

Specific Comments:

L34- Please provide reference(s) for physiological performance being influenced by hydration. I assume the physiological performance being influenced by temperature is specific to ectotherms but that is not clear in this sentence. It is also unclear if you mean both temperature and hydration (as the example you give suggests) or separately and combined. Please clarify.

We included “Ectothermic animals” in the sentence and clarified, providing the requested references:

“Also, while dehydration may alter ectotherms’ growth rates and behavior [10], thermal and hydric stress combined may decrease reproduction and survival [11].”

L35- I should be capitalized and i.e. should be in italics as it is latin

Done

L37-39: reword this sentence. Metabolic rates can definitely increase but EWL isn’t necessarily as closely related in squamates, or even across ectotherms. This might be specific to tropical forest species, if so this needs to be stated clearly.

We changed our sentence, adding a “can” to ease the tone: “As body temperature rises, evaporative body water losses can increase exponentially [8], like metabolic costs [9].”

L39: remove this generalization with ecotherms because it simply isn’t true. There are numerous studies demonstrating species resilience when faced with higher temperatures and dehydration.

Since the references in the original version demonstrate that heat stress and dehydration may have a significant impact on growth rates, reproduction, and survival, we see no problem with this sentence. We are not generalizing that all ectotherms will have their growth stunted by thermal stress and dehydration; we are merely stating that it is possible. Thus, we stated: “Also, when combined, heat stress and dehydration may strongly impair ectotherms’ growth rates, reproduction, and survival [10,11]. “

We did not refer to “resilience” in that track of the text, which has a different meaning in ecology.

L41- remove ‘such’ from this sentence

 removed

L43- this portion is confusing, please reword and clarify what you mean by animals “needing” higher temperatures. Do you mean they don’t start to pant or urinate until temperatures are higher than normal because they’re dehydrated?

 We followed Dr. Brusch’s advice:

“When ectothermic animals, as lizards, exhibit thermoregulatory behaviors, they likely integrate information on their hydration level (i.e. they actually thermohydroregulate) [12]. For instance, dehydrated lizards required exposure to higher temperatures to induce the onset of water-demanding thermoregulatory behaviors, such as panting or urinating [13–18]. Notwithstanding, these two behaviors may represent emergency responses (reviewed in Tattersall et al., [19]).”

L45- It is confusing to suddenly have an example with just thermoregulation in between sentences with thermohydroregulation. Please re-order.

 Please, see above.

L50- consider re-writing so that it reads “…water stress could help us understand how climatic restrictions may influence their activity and distribution.”

We considered it, but since “climatic restriction” is an influence of climate on activity and distribution, the combination of “climatic restriction may influence” seems redundant. 

L52- the previous sentence stresses the importance of considering both hydric and thermal constraints. But then this sentence emphasizes just a thermal trait measurement. Re-word this sentence to avoid conf”usion.

We changed the sentence to:

“The voluntary thermal maximum (VTmax) is a thermoregulatory trait that has been shown to integrate both thermal and water levels in frogs [21].”

L62- it is not clear, as written, the “them” that these typical thermoconformers are responding to. Oxygen conentrations? Typical thermal assays? Please clarify

Thanks, we replaced them by “heating rates and start temperatures”.

L64- change “would” to ‘could’

Changed.

L66- The saline injections needs to be clarified or removed. If you keep it, you need to define what you mean earlier in the introduction by dehydration. Often times it is simply referring to hyperosmolality unless you want to define it as hypernatremia, as is suggested by this reference/sentence.

We clarified: “Previous studies in desert lizards [14] have found no effects of dehydration and starvation on the VTmax, although this parameter may vary after the injection of saline solutions, a different way to alter plasma osmolality levels.”

L68- consider removing “magnitude and” to make this sentence crystal clear.

Thanks, done.

L69-71- this just means that some species VTmax is at or very close to their CTmax L72-76- you need to at least address species specific VTmax versus CTmax. It’s exclusion here is confusing. Especially considering the next paragraph starts with thermal safety margins and a mention of CTmax. I understand not taking up too much space in this paper for brevity but your justification (L69-71) for species living in potentially dangerously warm environments seems like cherry picking.

This is not the case for Chalcides ocellatus, which, despite being quite close to its CTmax holds up 24h; this species is now included as an example. CTmax is unknown for most species with survival time at VTmax, which are very few. So, that would be guessing upon the multiple factors that may increase the time for survival for some species compared to others, and that deserves a full study in itself. Also, if there were any other data on survival time at the VTmax for other species that changed the story, we would gladly add it here. We think the message here is clear, the Vtmax may induce function loss and death in a matter of hours.

L86- Please define or explain what you mean by ‘dry-skinned ectotherms’ I assume snakes and lizards

We clarified: “squamates, terrestrial arthropods”

L87- change to “might bury themselves within”

Thanks, done.

L129-130 AND L138-139: I would prefer if you included descriptive stats here. However, they can also be found in Table S2, which is okay. If you don’t include descriptive statistics here, please at least include something like “(all p<0.05)” since you use ‘significantly affected by’

We included the suggested “(all p<0.05)”

L132 and throughout- include units for the mean body mass, I assume 0.694 g. (also on L134 an a few other places in the results).

Thanks for pointing that, done.

L151- what are negative thermal margins? I can’t find in the methods what this means, please explain somewhere in the manuscript.

We synonymized “Thermal margins” with warming tolerance margins, which were explained in the methods.

L168-170- as written this is confusing, consider re-writing so that it reads “The extent and duration of thermally stressful events, and the percentage of lizard populations affected by them, increased by several orders of magnitude as VTmax decreased, and were associated with dehydration more than any other parameter.”

Thanks, replaced.

L174- change to “had a larger affect”

Changed.

L176- change to ‘changes in mass associated with dehydration”

Changed.

L176- I assume this is in reference to hydration in animal who are 10cm deep and not dehydration in general? Please make sure this is clear here.

No, burying 10cm deep and being dehydrated or not are independent conditions included in the models that generate the maps.

We rephrased to: “Neither the ability to bury down to 10 cm nor changes in mass associated with dehydration did not observably affect our thermal risk metrics.”

L196- what is meant by internal environment?

We wrote: They attributed these changes to internal rhythms or rhythms of the external environment. 

We meant to separate internally set rhythms and changes in the environment that surround the animal. We think that what the authors wrote is transcribed to the best of our possibilities.

L207- what is the “receptively” about? If the citations, then it needs to be within the brackets.

We moved “respectively” to within the brackets.

L209-219- unless you have tropical/temperature lizard-specific references here, remove metabolic water as that is typically negligible in reptiles except in some cool examples of desert herbivores/insectivores.

Lowering temperature and hence metabolic rate necessarily lowers metabolic water production. Therefore, even though it has yet to be determined whether or not metabolic water production is essential for survival across lizard species, the argument is still sound.

To address Dr. Brusch’s concerns, we clarified: “However, the absolute performance of traits, like locomotion, hearing, digestive and immune efficiency, and the rates of metabolic water production (ex. in desert-adapted reptiles), can decline as well (reviewed in [9,12,49]).”

L216-240- It is unclear what this paragraph adds to the discussion. This paragraph details studies in other species (large, long-lived desert lizards) that are very different from the two tropical species used in your experiment (small, short-lived leaf litter lizards). Consider removing it entirely for brevity.

This is a significant portion of the discussion with which the other reviewer has completely concurred with all that has been said. We believe that it gives our findings a more general application and points the way for further research. As a result, since the text is within the size requested for plos one articles, after considering it, we decided to retain it.

L226- Please see work by Hazard (e.g., 2001, PBZ) to update these sentences. Desert iguanas have salt glands which is not relevant to the two species your experiment used.

We updated the text as requested: “Still, desert iguanas did alter their VTmax in response to injections of concentrated saline solutions, leading to osmotic imbalance. Yet, desert iguanas have especially efficient ways to keep the ionic balance under heavy loses of body water, like salt glands, renal secretions, and water stored in their thick tails [14, 82].”

L245-246- osmotic pressure is measured in the plasma portion of blood. Similar to my comment on L66, it is unclear here the difference between hyperosmolality and ‘dehydrating processes’

We changed it to “temperate lizard species indicate that they maintained their maximal body temperatures either after experimental saline injections, or during dehydrating processes [15,16,23,43,47,55].”

L256- what is hydration rate? That is not mentioned anywhere in the manuscript and it is unclear what you are referencing. How fast they can absorb water in the GI tract? Water loss rates? Please clarify.

 We replaced “rate” by “level”.

L273- You did not measure plasma osmolality so this statement is not justified

We eased the tone and supplied a case study as follows:

“Based on the discussed lines of evidence, we propose that the VTmax of lizards, and possibly other ectothermic species, might be relatively more reactive to dehydration when the measured species: 1) presents less effective mechanisms for controlling plasma osmolality (e.g., salt glands),”

L274- the only known lizard with a water reservoir is the Gila Monster who can use their urinary bladder, similar to desert tortoises. Having a greater total amount of body water (because of being large bodied) does not equal a meaningful amount of great total body water as this value is proportional and typically 65-75% across terrestrial vertebrate taxa.

We clarified and added the reference for the gila monster: “This could allow them to maintain activity, with the resultant water ingestion, if available [54]. We hypothesize that such a strategy might be especially effective for organisms with special water reserves, such as the bladder (ex. the Gila monster [83]), and/or for species with low specific water loss rates (e.g., large desert lizards or tortoises [43,47,55]).”

L285- if you use quotations, please add a citation directly after so readers know who you are quoting here.

We removed the quotation marks because we are not citing anyone and provided a more in-depth explanation of the phrase: “However, interspecific variation in the magnitude of this thermal fear zone (i.e., between the VTmax and the CTmax), “

L322- change to “…who live over and within…”

This suggestion was deemed an error by the grammar checker, so we changed it to.: “Loxopholis ferreirai, is a semiaquatic and scansorial lizard species that lives over and within logs in flooded “Igapó” forests across the course of the Rio Negro and tributaries [60].”

L354- is this absolute humidity or relative humidity?

Thanks, we added “relative”.

L374- it is unclear what you’re trying to say here, all individuals, except two, stayed in the can? Please re-word for clarity.

Reworded:

“Before heating, lizards were left for a couple of minutes to assess whether they were willing to abandon the heating can or if they took it as a refuge. This procedure led to excerpt two individuals which were visibly unwilling to remain in it.”

change to “…tape wrapped around the side…”

Thanks, changed.

L407- change “starting temperature, heating rate, and body size.”

Done.

L408- change “Besides” to ‘additionally’

Done.

L410- change “a reviewer’s” to ‘potential’

Done.

---

## [Decision Letter · Decision Letter 3]

18 May 2023

Dehydration alters behavioral thermoregulation and the geography of climatic vulnerability in two Amazonian lizards.

PONE-D-21-29853R3

Dear Dr. Camacho,

We’re pleased to inform you that your manuscript has been judged scientifically suitable for publication and will be formally accepted for publication once it meets all outstanding technical requirements.

Kind regards,

Daniel de Paiva Silva, Ph.D.

Academic Editor

PLOS ONE

Additional Editor Comments (optional):

Dear Dr. Camacho,

I am pleased to inform you that your manuscript has been accepted for publication in PLoS One! Congratulations!

Daniel Silva, PhD

Reviewers' comments:

Reviewer's Responses to Questions

**Comments to the Author**

1. If the authors have adequately addressed your comments raised in a previous round of review and you feel that this manuscript is now acceptable for publication, you may indicate that here to bypass the “Comments to the Author” section, enter your conflict of interest statement in the “Confidential to Editor” section, and submit your "Accept" recommendation.

Reviewer #4: All comments have been addressed

2. Is the manuscript technically sound, and do the data support the conclusions?

Reviewer #4: Yes

3. Has the statistical analysis been performed appropriately and rigorously? 

Reviewer #4: Yes

4. Have the authors made all data underlying the findings in their manuscript fully available?

Reviewer #4: Yes

5. Is the manuscript presented in an intelligible fashion and written in standard English?

Reviewer #4: Yes

6. Review Comments to the Author

Reviewer #4: The authors have done a great job responding to all of my comments. I really commend them for their patience going through 2-3 rounds of review, which I know can be brutal sometimes. Job well done!

7. PLOS authors have the option to publish the peer review history of their article (what does this mean?). If published, this will include your full peer review and any attached files.

Reviewer #4: **Yes: **George A Brusch IV

---

## [Editor Report · Acceptance letter]

2 Jun 2023

PONE-D-21-29853R3 

Dehydration alters behavioral thermoregulation and the geography of climatic vulnerability in two Amazonian lizards. 

Dear Dr. Camacho:

I'm pleased to inform you that your manuscript has been deemed suitable for publication in PLOS ONE. Congratulations! Your manuscript is now with our production department. 

Kind regards, 

on behalf of

Dr. Daniel de Paiva Silva 

Academic Editor

PLOS ONE